# Inferring circuit mechanisms from sparse neural recording and global perturbation in grid cells

John Widloski[1]*, Michael P Marder[2], Ila R Fiete[2,3]*

[1]Department of Psychology, The University of California, Berkeley, United States; [2]Department of Physics, The University of Texas, Austin, United States; [3]Center for Learning and Memory, The University of Texas, Austin, United States

**Abstract** A goal of systems neuroscience is to discover the circuit mechanisms underlying brain function. Despite experimental advances that enable circuit-wide neural recording, the problem remains open in part because solving the 'inverse problem' of inferring circuity and mechanism by merely observing activity is hard. In the grid cell system, we show through modeling that a technique based on global circuit perturbation and examination of a novel theoretical object called the *distribution of relative phase shifts (DRPS)* could reveal the mechanisms of a cortical circuit at unprecedented detail using extremely sparse neural recordings. We establish feasibility, showing that the method can discriminate between recurrent versus feedforward mechanisms and amongst various recurrent mechanisms using recordings from a handful of cells. The proposed strategy demonstrates that sparse recording coupled with simple perturbation can reveal more about circuit mechanism than can full knowledge of network activity or the synaptic connectivity matrix.

DOI: https://doi.org/10.7554/eLife.33503.001

*For correspondence:
johnwidloski@berkeley.edu (JW);
fiete@mail.clm.utexas.edu (IRF)

**Competing interests:** The authors declare that no competing interests exist.

## Introduction

In systems neuroscience we seek to discover how neural responses and complex functionality can emerge from the dynamical interactions of neurons in circuits. For instance, the circuit mechanisms that give rise to orientation tuning in primary visual cortex have been closely studied for the better part of a century (*Hubel and Wiesel, 1959*). Despite these efforts, arbitrating between between different candidate mechanisms has been difficult. Our experimental tools are typically observational: Neurons are recorded, often during a behavior, in increasing numbers today (*Dombeck et al., 2010*; *Ahrens et al., 2012*; *Ziv et al., 2013*; *Dunn et al., 2016*). Our theoretical models usually run in the 'forward' direction: We build hypothesized circuits to reproduce the observed activity data. Because there often is a many-to-one mapping from plausible models to neural activity, it is difficult to know which model more accurately describes the underlying system. For this reason, it remains unsettled whether – to return to a familiar example – orientation tuning arises mostly from selective feedforward summation of inputs or lateral interactions (*Rivlin-Etzion et al., 2012*; *Kim et al., 2014*; *Takemura et al., 2013*; *Ferster and Miller, 2000*; *Sompolinsky and Shapley, 1997*).

Here, we show that grid cells (*Hafting et al., 2005*) provide a unique opportunity to understand cortical circuit mechanism, when coupled with a novel approach for doing so. The promise of our approach lies in the fact that (1) it is not merely observational but rather relies on perturbation, and (2) it provides a novel theoretical measure (the 'distribution of relative phase shifts' or DRPS) along which several competing feedforward and recurrent grid cell models can be distinguished with the perturbative experiments.

The structure of grid cell responses – with their periodic tuning to 2D space – makes the system particularly amenable to dissection, as we will see below. Grid cells have already yielded insight into

their underpinnings: All cells with a common spatial tuning period remain confined to a single 2D manifold in activity space, and this manifold is invariant over time even when grid cell tuning curves deform as the animals are moved between novel and familiar environments (*Yoon et al., 2013*; *Fyhn et al., 2007*), as well as during REM and non-REM sleep (*Gardner et al., 2017*; *Trettel et al., 2017*). These findings imply the existence of a 2D *continuous attractor dynamics* within or feeding into the grid cell circuit.

Many models reproduce the spatially periodic responses of individual grid cells or groups of cells (*Fuhs and Touretzky, 2006*; *Burak and Fiete, 2006*; *McNaughton et al., 2006*; *Hasselmo et al., 2007*; *Burgess et al., 2007*; *Kropff and Treves, 2008*; *Guanella et al., 2007*; *Burak and Fiete, 2009*; *Welday et al., 2011*; *Dordek et al., 2016*). These include models in which the mechanism of grid tuning is a selective feedforward summation of spatially tuned responses (*Kropff and Treves, 2008*; *Dordek et al., 2016*; *Stachenfeld et al., 2017*), recurrent network architectures that lead to the stabilization of certain population patterns (*Fuhs and Touretzky, 2006*; *Burak and Fiete, 2006*; *Guanella et al., 2007*; *Burak and Fiete, 2009*; *Pastoll et al., 2013*; *Brecht et al., 2014*; *Widloski and Fiete, 2014*), the interference of temporally periodic signals in single cells (*Hasselmo et al., 2007*; *Burgess et al., 2007*), or a combination of some of these mechanisms (*Welday et al., 2011*; *Bush and Burgess, 2014*). They employ varying levels of mechanistic detail and make different assumptions about the inputs to the circuit. Because exclusively single-cell models lack the low-dimensional network-level dynamical constraints observed in grid cell modules (*Yoon et al., 2013*), and are further challenged by constraints from biophysical considerations (*Welinder et al., 2008*; *Remme et al., 2010*) and intracellular responses (*Domnisoru et al., 2013*; *Schmidt-Hieber and Häusser, 2013*), we do not further consider them here. The various recurrent network models (*Fuhs and Touretzky, 2006*; *Burak and Fiete, 2006*; *McNaughton et al., 2006*; *Guanella et al., 2007*; *Burak and Fiete, 2009*; *Brecht et al., 2014*) produce single neuron responses consistent with data and further predict the long-term, across-environment, and across-behavioral state cell–cell relationships found in the data (*Yoon et al., 2013*; *Fyhn et al., 2007*; *Gardner et al., 2017*; *Trettel et al., 2017*), but are indistinguishable on the basis of existing data and analyses. Here we examine ways to distinguish between a subset of grid cell models, specifically between the recurrent and feedforward models, and also between various recurrent network models. We call this subset of models our *candidate models*. Our goal is not to provide new models of grid cell activity, but rather to show, through theory and modeling, how the candidate models could be feasibly distinguished through experiment.

The candidate models form a diverse set, with differences that carry important implications for mechanism and for how the network could have developed from plasticity mechanisms. The candidates first broadly partition into recurrent and feedforward models, depending on whether the dynamics that originate spatial tuning and velocity integration are within (recurrent) or upstream (feedforward) of the grid cell layer. Recurrent models further partition on the basis of two key features: topology (periodic or not) and locality of connectivity (from local to global).

Among recurrent models, the first candidate models are *aperiodic* networks (*Figure 1a*) (*Burak and Fiete, 2009*; *Widloski and Fiete, 2014*): Network connectivity has no periodicity (flat, hole-free topology) and it is purely local (with respect to an appropriate or 'topographic' rearrangement of neurons only nearby neurons connect to each other). Despite the aperiodic and local structure of the network, activity in the cortical sheet is periodically patterned (under the same topographic arrangement). In this model, co-active cells in different activity bumps in the cortical sheet are not connected, implying that periodic activity is not mirrored by any periodicity in connectivity. Interestingly, this aperiodic network can generate spatially periodic tuning in single cells because, as the animal runs, the population pattern can flow in a corresponding direction and as existing bumps flow off the sheet, new bumps form at the network edges, their locations dictated by inhibitory influences from active neurons in other bumps (*Figure 1e*). From a developmental perspective, associative learning rules can create an aperiodic network (*Widloski and Fiete, 2014*), but only with the addition of a second constraint: Either that associative learning is halted as soon as the periodic pattern emerges, so that strongly correlated neurons in different activity neurons do not end up coupled to each other, or that the lateral coupling in the network is physically local, so that grid cells in the same network cannot become strongly coupled through associative learning even if they are highly correlated, because they are physically separated. In the latter case, the network would have to be topographically organized, a strong prediction.

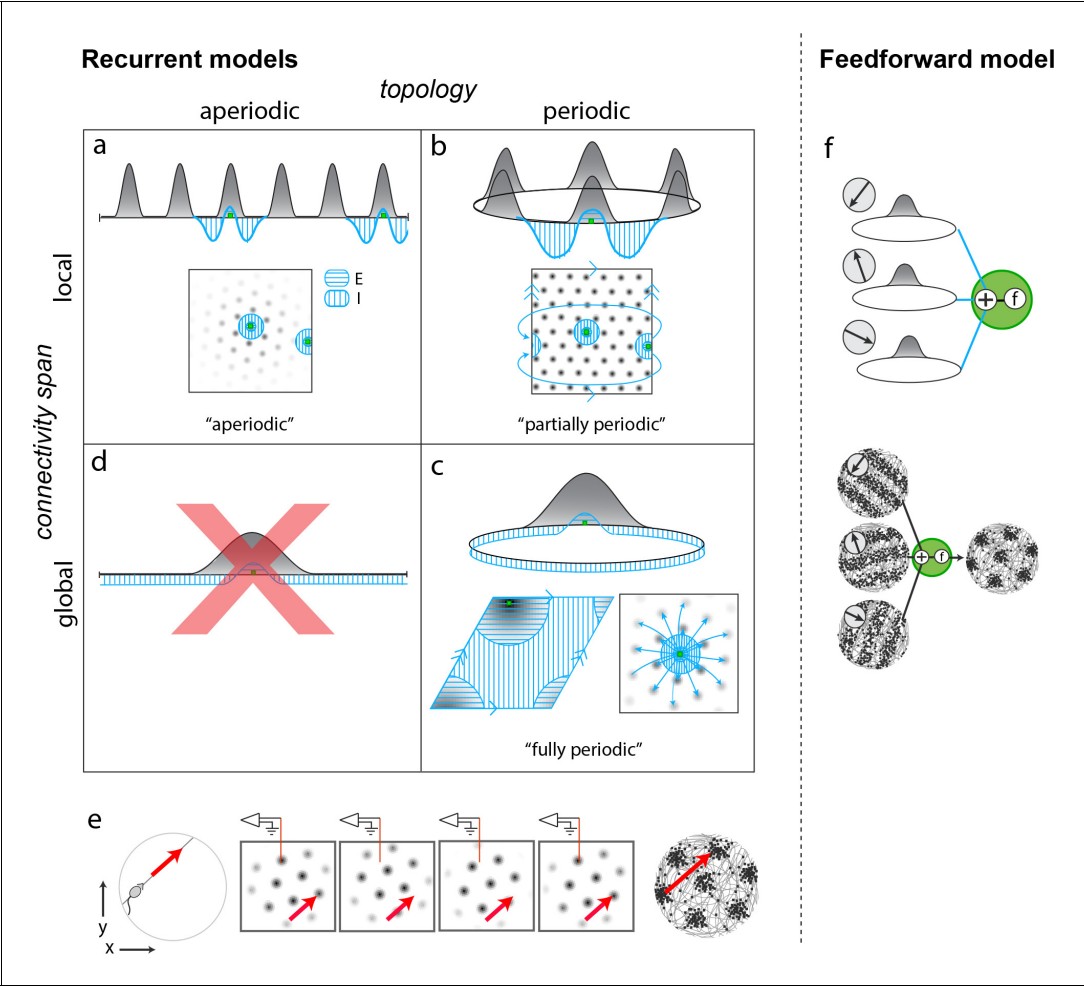

**Figure 1.** Mechanistically distinct models that cannot be ruled out with existing results. (**a–d**) Recurrent pattern-forming models. Gray bumps: population activity profiles. Blue: Profile of synaptic weights from a representative grid cell (green) to the rest of the network. Bottom of each panel: 2D network; Top: equivalent 1D toy network. Matching arrows along a pair of straight network edges signify that those edges are glued together. (**a**) Aperiodic network: (*Burak and Fiete, 2009*; *Widloski and Fiete, 2014*): local connectivity without periodic network boundaries. (**b**) Partially periodic network (*Burak and Fiete, 2009*): local connectivity in a network with periodic boundaries. (**c**) Bottom left and top: Fully periodic network (*Guanella et al., 2007*; *Burak and Fiete, 2006*; *Fuhs and Touretzky, 2006*; *Pastoll et al., 2013*; *Brecht et al., 2014*; *Widloski and Fiete, 2014*), with global connectivity and periodic boundaries. Bottom right: multi-bump network with local-looking connectivity but long-range connections between co-active cells in different bumps. This model is mathematically equivalent to a fully periodic model (see *Figure 1—figure supplement 1*). (**d**) A network with a single activity bump and without periodic boundaries cannot properly retain phase information as the bump moves around: it will not be a good integrator of animal velocity and is not a candidate mechanism. (**e**) Movement of the animal (left) causes a flow of the population pattern in proportion to animal velocity (four snapshots over time in center panels) for the models in (**a–c**). Red line: Electrode whose tip marks the location of a recorded cell. The recorded cell's response is spatially periodic (right; spikes in black), like grid cells. (**f**) Feedforward model: A grid cell (green) receives and combines inputs that are spatially tuned with uniform resolution across open spaces (implying these inputs reflect path integration-baed location estimates). These inputs may arise from recurrent ring attractor networks (*Mhatre et al., 2012*; *Blair et al., 2008*) (top) and exhibit stripe-like spatial tuning either in their firing rates (*Mhatre et al., 2012*) (bottom left) or firing *phase* with respect to the theta-band LFP oscillation (*Welday et al., 2011*; *Bush and Burgess, 2014*) (not shown). Or, they could arise from place cells assumed to path integrate (*Kropff and Treves, 2008*; *Dordek et al., 2016*). Selective feedforward summation followed by a nonlinearity produces grid-like responses (bottom right).

DOI: https://doi.org/10.7554/eLife.33503.002

The following figure supplements are available for figure 1:

**Figure supplement 1.** (Weakly) coupling neurons based on periodic activity patterning converts an aperiodic network into an effectively fully periodic one.

DOI: https://doi.org/10.7554/eLife.33503.003

**Figure supplement 2.** The a priori theoretical implausibility of partially periodic networks.

DOI: https://doi.org/10.7554/eLife.33503.004

Second are *fully periodic* networks (*Figure 1c*) (*Guanella et al., 2007*; *Fuhs and Touretzky, 2006*; *Pastoll et al., 2013*; *Brecht et al., 2014*). The network is topologically a torus with periodic boundary conditions between the pairs of opposite edges, and connectivity is global: There is no neural rearrangement under which network connectivity will be local. It is mathematically equivalent to view this network as having a single activity bump (*Burak and Fiete, 2006*; *Guanella et al., 2007*) or having multiple periodically arranged bumps with inter-bump connections (*Burak and Fiete, 2009*). In this network, periodic connectivity underlies periodic activity. Developmentally, a fully periodic network would naturally arise if associative plasticity continued post-pattern formation, so that the topology of activity and connectivity would come to mirror each other (*Widloski and Fiete, 2014*).

Third are *partially periodic* networks (*Burak and Fiete, 2009*) (*Figure 1b*) with periodic boundary conditions (torus topology) but only local connectivity on the torus after appropriate rearrangement of neurons. In this model, neural activity on the cortical sheet is multi-peaked and periodic (under appropriate rearrangement). Conceptually and developmentally, these networks are the strangest: None of the connectivity in the bulk of the network reflects the periodic nature of activity within it, except for the connectivity necessary to connect together neurons across the edges of an initially aperiodic sheet of cells. The wiring of this 'edge' subset of neurons must, unlike the rest of the cells, depend on details of the periodic activity pattern to make sure that opposite edge bumps are 'aligned' before joining (*Figure 1—figure supplement 2*). It is unclear how activity-dependent plasticity rules, which could wire together faraway edge neurons based on activity, would refrain from doing the same for the rest, to maintain otherwise local connectivity.

The fourth potential combination of topology and locality is not permitted: it is not possible to obtain grid-like activity from neurons with global connectivity (and single-bump activity) but aperiodic boundaries (topologically flat hole-free networks), *Figure 1d*.

*Feedforward* models of grid cell activity form a robust and growing set. In these models, grid cells merely sum and transform with a pointwise nonlinearity inputs that are already spatially tuned with roughly uniform coverage and resolution across the environment (*Figure 1f*) (*Kropff and Treves, 2008*; *Welday et al., 2011*; *Mhatre et al., 2012*; *Bush and Burgess, 2014*; *Hasselmo and Brandon, 2012*; *Dordek et al., 2016*; *Stachenfeld et al., 2017*); thus, it is implicitly assumed that the upstream inputs to grid cells have already performed path integration. These feedforward models, which we propose could be distinguished from recurrent models with the proposed perturbative approach, themselves segment into two major varieties. One type (*Welday et al., 2011*; *Mhatre et al., 2012*; *Bush and Burgess, 2014*; *Hasselmo and Brandon, 2012*) generates low-dimensional grid cell population activity across environments (e.g., in *Welday et al., 2011*, three upstream circuits, each a 1D continuous attractor network, integrate one component of animal velocity aligned to each of the three primary directions of a triangular lattice; the combined response is 2-dimensional, and preserved across environments; other models of this type differ in details but are similar in this regard), as predicted also by the recurrent models and found in the data (*Yoon et al., 2013*). In the second type (*Kropff and Treves, 2008*; *Dordek et al., 2016*; *Stachenfeld et al., 2017*), the grid cell pattern for an environment depends on the place cell pattern for that environment. Thus, when the place cell representations remap across environments, the model grid cells will not preserve their spatial relationships.

Our candidate models are the set of recurrent and feedforward models described and cited above. They are architecturally and mechanistically distinct in ways both large and subtle: they differ in whether grid cells or their upstream inputs are performing velocity-to-location integration, in whether spatial patterning originates wholly or only partly within grid cells, and in the structure of their recurrent circuitry. As already noted, some of the subtle-seeming structural differences have important implications for circuit development: different connectivity profiles and topologies require distinct models of plasticity and experience during circuit formation (*Widloski and Fiete, 2014*). Nevertheless, candidate recurrent and feedforward models that exhibit approximate 2D continuous attractor dynamics are difficult to distinguish on the basis of existing data.

As we discuss at the end, neither complete single neuron-resolution activity records nor complete single synapse-resolution weight matrices (connectomes) will fully suffice to distinguish between the candidate models because they are observational or correlative techniques: they do not probe the causal origin of the observed responses.

We show how it is nevertheless possible to gain surprisingly detailed information about the grid cell circuit from a feasible perturbation-based experimental strategy, enough to discriminate between the candidate models.

## Results

### A perturbation-based probe of circuit architecture

The question of mechanism is focused on a pre-specified set of neurons or local circuit: Is the observed low-dimensional grid cell activity primarily based on recurrent interactions within the set, and how, or is it inherited from feedforward drive originating outside this set? We refer to a perturbation as simple, low-dimensional and *global* in this context if it affects most cells within this set without regard to their individual functional identities, and does not affect those outside. In what follows, we consider the set to consist of all grid cells and conjunctive cells in one (or more) grid modules (*Stensola et al., 2012*), as well as the interneurons that surround them; toward the end we discuss the effects of perturbing subpopulations or bigger sets.

The central idea is as follows: Globally perturbing either the time-constant of neurons or the gain of recurrent inhibition is predicted to affect cell–cell spatial tuning relationships in candidate models in a specific way that can be robustly observed and characterized from ultra-sparse sampling of neurons in the network, and the predicted effects differ across candidate mechanisms.

To present the idea, we consider a thought experiment on the aperiodic recurrent network models. We will retake the larger perspective, of discriminating between the various model categories, immediately afterward. In aperiodic models, perturbing the gain of recurrent inhibition or the time-constant of neurons will induce a shift in the period of the internal population pattern (*Figure 2—figure supplement 1*). Let us quantify the change in period by the population period stretch factor, $\alpha = |\frac{\lambda_{pop,post}}{\lambda_{pop,pre}} - 1|$ (where $\lambda_{pop,pre}$ is the pre-perturbation population period). Without loss of generality, suppose that the focus of pattern expansion is at the left edge, *Figure 2a* (blue: original pattern, red: expanded pattern). Each neuron can be assigned a *population phase* with respect to the period of the population pattern: If the phase at the left edge is called 0 (again without loss of generality), neurons lying at integer multiples of the original period also originally had a phase of 0 (*Figure 2b*). However post-expansion, the population phase of a neuron originally one period away from the left edge is no longer zero (*Figure 2a,b*). Let us call the shift in the population phase of this neuron one 'quantum' (*Figure 2b*), and denote it by $\Delta$. The quantum of shift must be $\Delta = |1 - \frac{\lambda_{pop,pre}}{\lambda_{pop,post}}| = \frac{\alpha}{1+\alpha}$ ($\approx \alpha$ for small perturbations). A neuron $K$ periods away will shift in phase by $K$ quanta, *Figure 2b*. If there are $M$ bumps in the population pattern, the largest shift will be $M$ quanta, or $M\Delta$ (modulo 1). If we construct a distribution of shifts in population phase pre- to post-expansion for cells across the network, the distribution will be quantal, with $2M$ peaks (assuming the biggest phase shift, $M\Delta$, is less than $1/2$, because phase is a periodic variable that we parameterize as running between $-1/2$ to $1/2$; this condition can be met by keeping the perturbation small, such that $\Delta < 1/(2M)$), *Figure 2—figure supplement 2* and *Figure 2c*. In other words, for small perturbations, the number of peaks in this distribution is predicted to be twice the number of bumps in the original population pattern. We will call this distribution of relative phase shifts the DRPS.

Practically, however, the grid cell network might not be topographically well-ordered on a sufficiently fine scale (*Heys et al., 2014*), and one cannot simply image the population response and expect to read off pattern phases for each cell as in *Figure 2a,b*.

Fortunately, the distribution of shifts in the difficult-to-observe *population phases* of cells, based on instantaneous and topographically ordered population activity snapshots, is mirrored in the distribution of shifts in the *relative phase* of the straightforward-to-observe and time-averaged *spatial tuning curves* of cells (*Figure 2d*). Consider a pair of cells one population period apart pre-perturbation, so they have the same population phase (circle, square or square, triangle in *Figure 2a*). These cells are co-active and have the same spatial tuning curves, and thus a *relative spatial tuning phase* of 0 (circle, square or square, triangle in *Figure 2d*, top). Post-perturbation, their spatial tuning curves will be shifted relative to each other by the same amount as the shift in their individual population phases (circle, square or square, triangle in *Figure 2d*, bottom). In other words, cells one bump apart in the original population pattern will exhibit one quantum of shift in their relative spatial

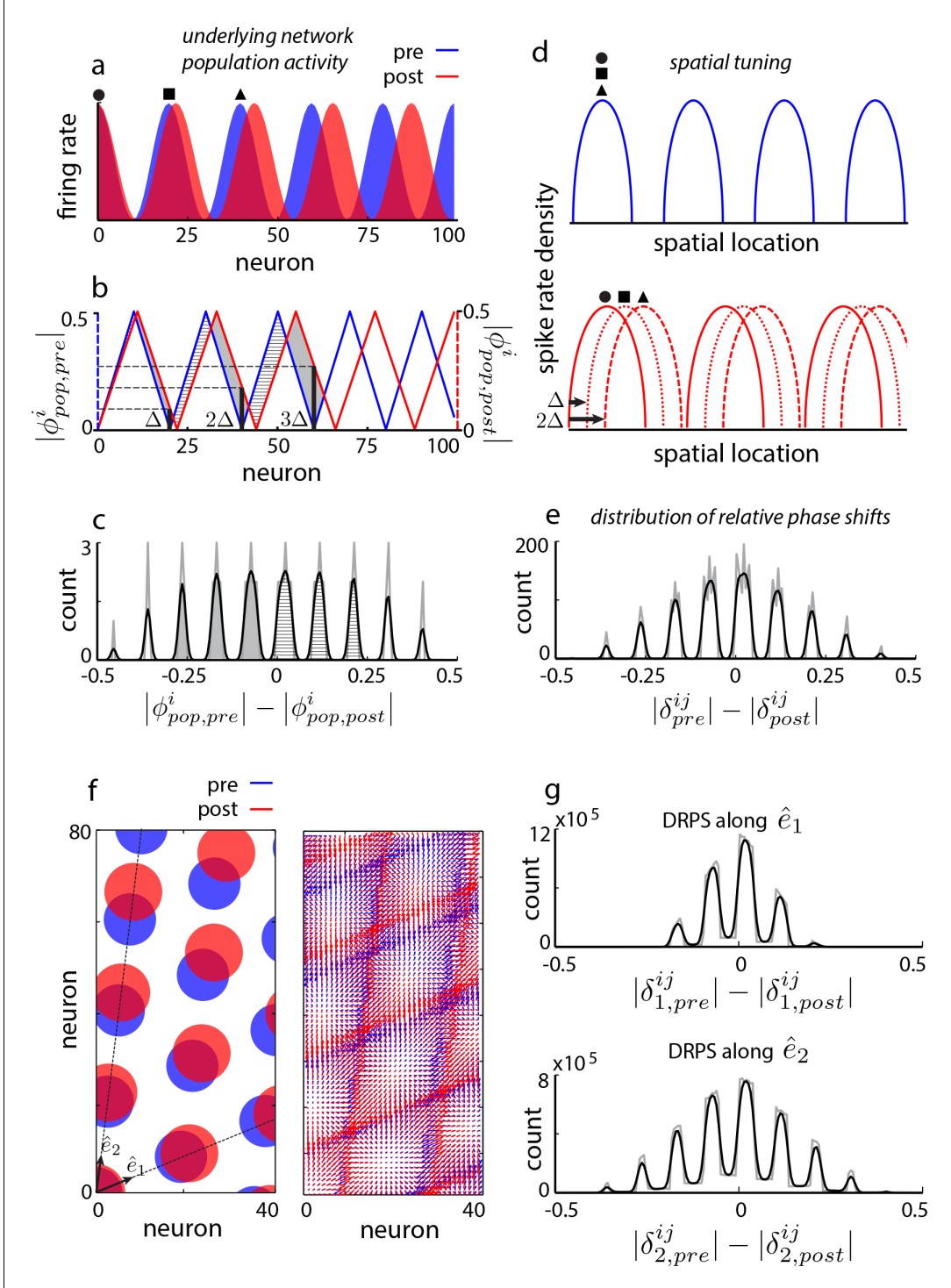

**Figure 2.** Global perturbation with analysis of phase shifts: signatures of recurrent patterning. (a) Schematic of population activity before (blue) and after (red) a 10% period expansion ($\alpha = 0.1$; the center of expansion is shown at left, but results are independent of this choice) in an aperiodic network. Circle, square, triangle: three sample cells with the same pre-expansion population phase. (b) The *population phase* $\phi_{pop}^i$ of the *i*th neuron is defined as $\phi_{pop}^i = ((i-1)/\lambda_{pop}) \bmod 1$ where $\lambda_{pop}$ is the population pattern period. Plotted: population phase magnitudes pre- (blue) and post- (red) expansion (phase magnitude is given by the Lee distance, $|\phi| = \min(||\phi||, 1 - ||\phi||)$, where $|| \cdot ||$ denotes absolute value). (c) Histogram of quantal shifts in the pre- to post-expansion population phases for all (n = 100) cells in the network. Gray line: raw histogram (200 bins). Black line: smoothed histogram (convolution with 2-bin Gaussian). Negative (positive) phase shifts arise from gray-shaded (horizontally-striped) areas in (b). (d–e) Quantal shifts in the population phase (experimentally inaccessible) are mirrored in shifts in the pairwise relative phase of spatial tuning between cells (experimentally

*Figure 2 continued on next page*

*Figure 2 continued*

observable). (**d**) Schematic of spatial tuning curves of three cells (circle, square, triangle) from (**a**). Pre-expansion the tuning curves have the same phase (top), thus their relative spatial tuning phases are zero. The tuning curves become offset post-expansion (bottom), because the shift in the population pattern forces them to stop being coactive. (**e**) Distribution of relative phase shifts (DRPS; gray). Relative phase between cells $i, j$ ($d^{ij}$ is the offset of the central peak in the cross-correlation of their spatial tuning curves; $\lambda$ is their shared spatial tuning period). A relative phase shift is the difference in relative phase between a pair of cells pre- to post-perturbation. Black: smoothed version. There are (100 choose 2)=4950 pairwise relative phase samples. (**f**) Population activity pattern and pattern phase pre- and post-expansion in a 2D grid network (as in (**a–b**)). Dotted lines: principal axes of the population pattern (left). An arrow marks each cell's population phase (right). (**g**) DRPS for the two components of 2D relative phase (as in (**e**); see Materials and methods). Samples: (3200 choose 2).

DOI: https://doi.org/10.7554/eLife.33503.005

The following figure supplements are available for figure 2:

**Figure supplement 1.** Dynamical simulations of the aperiodic network with LNP dynamics: gradual change in population period.

DOI: https://doi.org/10.7554/eLife.33503.006

**Figure supplement 2.** When the 2:1 relationship between number of peaks in the DRPS and the number of bumps in the population pattern breaks down.

DOI: https://doi.org/10.7554/eLife.33503.007

**Figure supplement 3.** Alternative formulation of the DRPS.

DOI: https://doi.org/10.7554/eLife.33503.008

**Figure supplement 4.** Cortical Hodgkin-Huxley (CHH) simulations to assess the effects of cooling as an experimental perturbation and to elucidate the link between temperature and parameter settings in grid cell models with simpler neurons.

DOI: https://doi.org/10.7554/eLife.33503.009

tuning. The relative phase of spatial tuning for a pair originally separated by $K$ periods will shift by $K$ quanta post-perturbation (e.g., circle, triangle in *Figure 2d*, bottom: spatial tuning curves shift by two quanta in phase because these cells were two periods apart in the original population pattern).

This series of theoretical observations leads us to construct a predicted *distribution of relative phase shifts* (DRPS) from all pairs of neurons, *Figure 2e*. The DRPS is quantal and has the same number of peaks as the distribution of shifts in population phase (*Figure 2c*). Indeed, multiplying the number of peaks in the multimodal DRPS by $1/2$ gives the number of bumps in the original population pattern, if the quantal shift size is sufficiently small. The DRPS is a property of patterning in an abstract space, independent of how neurons are actually arranged in the cortical sheet. It can be obtained from the spatial tuning curves of cells recorded simultaneously through either conventional electrophysiology or imaging. As we show later, a robust estimate of the full DRPS can be obtained from only a handful of cells.

In 2D, relative phase is a vector. The two components are each computed simply as in 1D, along each of the two principal axes of the spatial tuning grid. For an aperiodic network, for small enough perturbations, the total number of bumps in the population pattern can be inferred to be a quarter of the product of the number of peaks in the two relative phase shift distributions from the two components of the relative phase (*Figure 2f–h*).

## Experimental knobs to modulate the population pattern

To generate the DRPS in experiment and use it to distinguish between grid cell models requires an experimental knob that can be turned to change the period of the population activity pattern. Temperature is one potential knob: Cooling a biological system reduces reaction rates and increases time-constants through the Arrhenius effect (*Katz and Miledi, 1965*; *Thompson et al., 1985*; *Moser and Andersen, 1994*; *Long and Fee, 2008*). However, existing models of grid cells are based on simplified rate-based or linear-nonlinear Poisson (LNP) spiking units, and it is unclear which parameters to modify to correctly predict the effects of cooling the neural circuit: Varying a 'neural' time-constant parameter in a recurrent network of simple units may or may not change the population pattern period, depending on whether PSP height is scaled together with the time-constant change (*Widloski and Fiete, 2014*; *Beed et al., 2013*) or not. To better predict the effects of cooling on grid cell period, we constructed more detailed grid cell models using cortical Hodgkin-Huxley model neurons (*Pospischil et al., 2008*) whose parameters accommodate thermal effects (*Hodgkin et al., 1952*; *Katz and Miledi, 1965*).

The population period in aperiodic grid cell models built from Hodgkin-Huxley neurons is pulled in opposing directions by temperature modulations in ion-channel biophysics and synaptic signalling (*Figure 2—figure supplement 4*). However, the dominant influence on network response comes from the growth in the PSP time-constant with cooling and results in an overall expansion of the population period (*Figure 2—figure supplement 4*).

This result allows us to conclude that the net effect of cooling the biological circuit should be an expansion in the period, if the circuit is recurrently connected and aperiodic. It also allows us to continue using simple rate-based and LNP spiking models because we can now interpret how to scale parameters as a function of temperature: It is most appropriate to scale the time-constant inversely with temperature, while essentially keeping the PSP height fixed (*Figure 2—figure supplement 4*).

The strength of recurrent inhibition is another experimental knob. Unlike temperature, manipulating the gain of inhibitory synaptic conductances has a relatively unambiguous interpretation in grid cell models. Experimentally, the strength of inhibition might be modulated by locally infusing allosteric modulators that increase inhibitory channel conductances (e.g. benzodiazipines; *Rudolph and Möhler, 2004* and personal communication with C. Barry). In both cortical Hodgkin-Huxley based models grid cell models (*Figure 2—figure supplement 4*) and rate-based models, a gain change in inhibitory conductances predicts a change in the period of the population pattern (*Figure 2—figure supplement 4* and *Moser et al., 2014*, *Widloski and Fiete, 2014*).

To summarize, thermal perturbation (cortical cooling) and biochemical perturbation (drug infusions to alter the gain of recurrent inhibition) are two experimental manipulations that could, according to the models, alter the period of a recurrently formed population pattern and thus may act as appropriate knobs to enable the construction of the DRPS.

## Discriminating among recurrent architectures

In dynamical simulations of the various plausible candidates (Materials and methods), the same global perturbations have different effects, resulting in distinct predicted DRPS's. To generate maximally robust and easy-to-interpret predictions, we focus on how the candidate models differ with respect to one simple property of the DRPS: the overall width of its envelope.

In aperiodic networks (*Figure 1a*) with smooth boundaries for accurate integration (*Burak and Fiete, 2009*), an incremental increase in the strength of global perturbation results in incremental expansion of the population activity pattern (*Figure 3a*, red, and *Figure 2—figure supplement 1*) (see *Widloski, 2015* for an analysis of boundary conditions and permitted number of peaks). Thus, the peaks in the DRPS will incrementally spread out, producing a DRPS envelope that gradually and smoothly widens with perturbation strength (*Figure 3b–c*, red). In addition, because the change in period is incremental when the perturbation strength is gradually increased, it may be possible to estimate the number of bumps in the population pattern by counting peaks in the DRPS.

Partially periodic networks (*Figure 1c*), unlike aperiodic networks, must because of their symmetry accommodate an integer number of complete activity bumps in a way that is perfectly periodic (*Widloski, 2015*). The bumps and spacings within a partially periodic network are identical and the population pattern (if the geometry of the 2D pattern is fixed) is characterized simply by the number of bumps, which is constrained to be an integer. Incrementally increasing the perturbation strength is thus predicted to first result in no change, followed by a sudden change when the network can accommodate an entire additional bump, *Figure 3a* (purple) (or an additional row of bumps in 2D, assuming the pattern does not rotate as a result of the perturbation; see Discussion). As soon as a new bump has been inserted into the population pattern, the phase shifts will be large even for cells in adjacent bumps, and the DRPS will be wide. To summarize, for partially periodic networks, incremental changes in perturbation strength are therefore predicted to result in a stepwise (stepping to maximal width) change in the DRPS (*Figure 3b–d*, purple).

Counting peaks to estimate the number of bumps in the underlying population pattern after a stepwise change in the DRPS will likely result in substantial underestimation: because the phases shift by a large step when a change occurs, if a shift of $M$ quanta already exceeds one cycle, the DRPS will not distinguish between an $M$-bump and a $K$-bump network ($K>M$; *Figure 2—figure supplement 2* and e.g., *Figure 3b*: compare peaks in the solid and dashed lines for small and large perturbations, respectively).

Finally, in the fully periodic network (*Figure 1b*) the globally periodic connectivity completely determines the population period of the pattern, and changes in the neural time-constants or

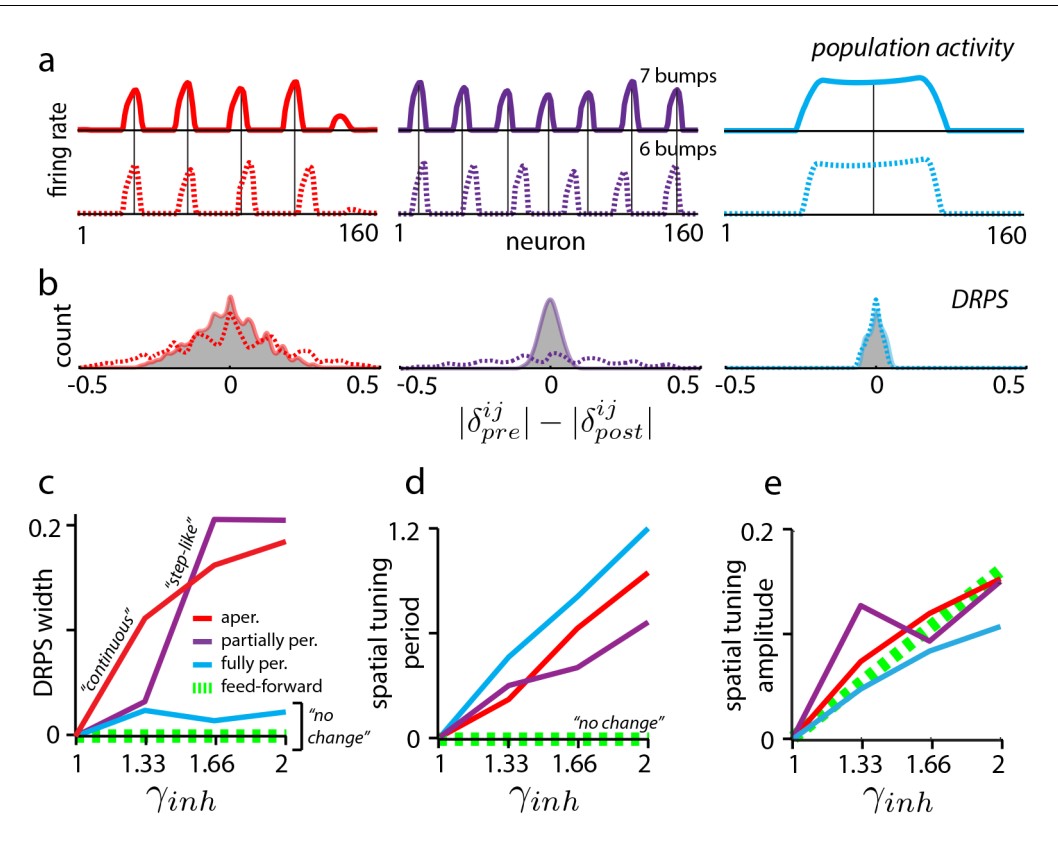

**Figure 3.** Effects of perturbation in recurrent and feedforward neural network simulations: predictions for experiment. (**a**) Simulations of aperiodic (column 1), partially periodic (column 2), and fully periodic (column 3) networks show changes in the population pattern pre-perturbation (first row; $\gamma_{inh} = 1$) to post-perturbation (second row; $\gamma_{inh} = 1.33$). Solid vertical lines: pre-perturbation bump locations. (Simulation details in Materials and methods.) (**b**) Perturbation-induced DRPS in the various networks for two perturbation strengths ($\gamma_{inh} = 1.33$: solid line and filled gray area; $\gamma_{inh} = 1.66$: dotted line), both relative to the unperturbed case. (**c**) DRPS width ($\sigma_{DRPS}$, defined as the standard deviation of the DRPS) as a function of perturbation strength for the different networks. Dashed green line: feedforward networks (predicted, not from simulation). The step-like shape for the partially periodic network is generic; however, the point at which the step occurs may vary from trial to trial. (**d–e**) Change in spatial tuning period (**d**) and amplitude (**e**) as a function of the perturbation strength (see Materials and methods). Change is defined as $|\frac{X_{post}}{X_{pre}} - 1|$, where $X$ is the spatial tuning period or amplitude.

DOI: https://doi.org/10.7554/eLife.33503.010

The following figure supplement is available for figure 3:

**Figure supplement 1.** Changes in spatial tuning period in neural network simulations of the grid cell circuit are due to changes in both the population period and the velocity response of the network.
DOI: https://doi.org/10.7554/eLife.33503.011

network inhibition strength do not alter it (*Figure 3a*, blue). Thus, the same global perturbations that effected changes in the population period in the other recurrent models (*Figure 3a*, red and purple) have no effect in the fully periodic network. The DRPS is consequently predicted to remain narrow, unimodal, and peaked at zero (*Figure 3b–c*, blue).

## Discriminating feedforward from recurrent architectures

If low-dimensional dynamics and spatially tuned responses first originate upstream of the perturbed set, then the perturbations will leave unchanged the spatial tuning phases of grid cells, preserving grid cell–grid cell relationships. This prediction holds even if grid cells play a role in constructing their particular patterns of spatial tuning, for instance by combining elements that are already spatially tuned as when stripe-tuned inputs are combined to generate 2D lattice responses

(*Mhatre et al., 2012*; *Welday et al., 2011*; *Bush and Burgess, 2014*) (*Figure 1f*). Thus, for feedforward models, as for fully periodic (recurrent) networks, the DRPS is predicted to remain narrowly peaked at zero across a range of perturbation strengths (*Figure 3c*, dashed green line).

Further, perturbing grid cells but not their spatially patterned feedforward inputs will not affect their spatial tuning. By contrast, in all recurrent models (*Figure 1a–c*), perturbing the grid cell network induces a change in the efficacy with which feedforward velocity inputs drive the population phase over time, thus the spatial tuning period of cells is predicted to change even if the population period does not (as in fully periodic networks – see *Figure 3—figure supplement 1*), *Figure 3d*. This expansion in spatial tuning period with global perturbation strength is predicted to hold for all three recurrent network classes, and distinguishes fully periodic recurrent networks from feedforward ones.

Finally, in both feedforward and recurrent neural network models, the amplitude of the grid cell response will change in response to perturbation (*Figure 3e*). This universal prediction of amplitude change with perturbation can be used as an assay of whether the attempted global perturbation is in effect.

## Data limitations and robustness

We consider two key data limitations. First, it is not yet experimentally feasible to record from all or even a large fraction of cells in a grid module. Interestingly, the proposed method is tolerant to extreme sub-sampling of the population: a tiny random fraction grid cells from the population (10 out of e.g. 1600 cells, or 0.6%) can capture the essential structure of the full DRPS, *Figure 4a*, including its overall width and the detailed locations of its multiple peaks. This robustness to subsampling is dramatically better than in statistical inference methods, where even 'sparse' methods can require $\sim 2$ orders of magnitude denser data (*Soudry et al., 2015*).

The second limitation arises from the limited accuracy with which spatial tuning and relative phase can be estimated from finite data. In tests that depend only on the width of the DRPS (e.g. *Figure 3*), this phase uncertainty is not a serious limitation.

Resolving the relative phase accurately becomes important when counting DRPS peaks to estimate how many bumps are in the underlying population pattern of a recurrent network. The spacing between DRPS peaks determines the required tolerance in relative phase (*Figure 4—figure supplement 1*). DRPS peak spacing (in the aperiodic network) increases with the stretch factor at small stretch factors (*Figure 4b* and *Figure 4—figure supplement 1*), but the stretch factor must still obey $\alpha \approx \Delta < 1/(2M)$ (where $M$ equals the larger of the number of bumps along the two dimensions of the population pattern; *Figure 4b*) to avoid underestimating the number of bumps in the population pattern.

Fortunately, it is possible to gain progressively better estimates of relative phase over time even if there is substantial drift in the spatial responses of cells, because relative phases remain stable in a fixed network (*Yoon et al., 2013*) (here 'fixed' means that a given perturbation strength is stably maintained). Many estimates of relative phase may be made from short pieces of the trajectory, and these estimates averaged together (similar to the methods used in *Yoon et al., 2013* and *Bonnevie et al., 2013*).

To distinguish $M = 5$ bumps per dimension based on structure within the DRPS requires a stretch factor $\alpha \approx \Delta < 1/(2M) = 0.1$, and a phase noise of 0.02 or smaller (*Figure 4—figure supplement 1*), which would require an approximately 8 min recording (estimated from grid cell and trajectory data, http://www.ntnu.edu/kavli/research.grid-cell-data), *Figure 4c*. Distinguishing seven bumps would require $\alpha \leq 0.07$, phase noise less than $0.01$, and a 35 min recording.

In summary, the proposed method has high tolerance to subsampling and more limited tolerance to phase uncertainty, which can be reduced by averaging estimates over time.

## A decision tree for experimental design

We lay out a decision tree with an experimental workflow for discriminating between disparate feedforward and recurrent grid cell mechanisms, all of which exhibit approximate 2D continuous attractor dynamics at the population level (*Figure 5*).

We start with the 'specific' approach, which, according to our model, has more discriminatory power than the 'nonspecific approach' described later. The experimental demands of this approach

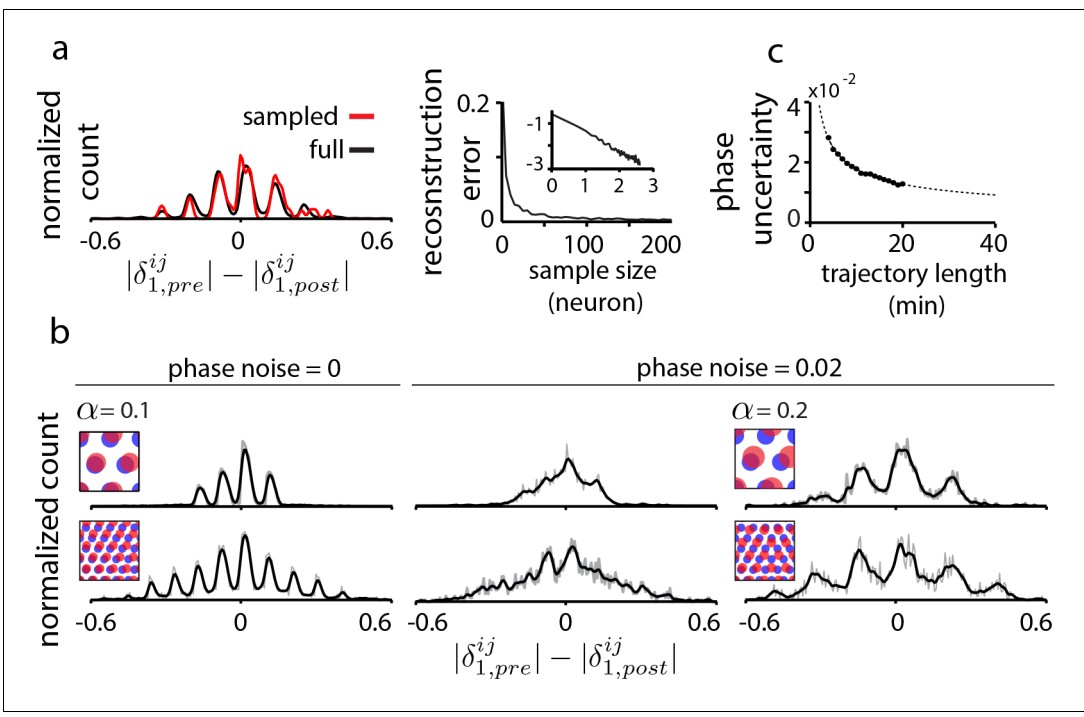

**Figure 4.** Data limitations and the resolvability of predictions. (**a**) Left: The quantal structure of the DRPS (along first principal axis of the 2D phase) is apparent even in small samples of the population (black: full population; red: n = 10 cells out of 1600; stretch factor $\alpha = 0.1$). Right: The L2-norm difference between the full and sampled DRPS as a function of number of sampled cells. Inset: log-log scale. (**b**) First and second rows: DRPS for population patterns with different numbers of bumps (gray line: raw; black line: smoothed with 2-bin Gaussian). Column 1: zero error or noise in estimating relative phase. Column 2: same DRPS' as in column 1, but with phase estimation errors (i.i.d. additive Gaussian noise with zero mean and standard deviation 0.02 for each component of the relative phase vector, $\vec{\delta}^{ij}$). Column 3: Increasing the stretch factor ($\alpha = 0.2$) renders the peaks in the DRPS more discernible at a fixed level of phase noise. For the 5-bump pattern (second row), $M\Delta \approx M\alpha = 5 \times 0.2 > 1/2$ and thus the number of peaks in the DRPS times 1/2 at this larger stretch factor will underestimate the number of bumps in the underlying population pattern. (**c**) In grid cell recordings (data from **Hafting et al., 2005**), the uncertainty in measuring relative phase, as estimated by bootstrap sampling from the full dataset (see Materials and methods), declines with the length of the data record according to $T^{-\frac{1}{2}}$ (dotted line). *Parameters*: $\lambda_{pop,pre} = 40/3 \approx 13.3$ neurons (a) =20 neurons (b, top row),=8 neurons (b, bottom row); $\alpha = 0.1$; $\hat{e}_1 = [1, 0]$; network size: $40 \times 40$ neurons.
DOI: https://doi.org/10.7554/eLife.33503.012

The following figure supplement is available for figure 4:

**Figure supplement 1.** Effects of uncertainty in phase estimation.
DOI: https://doi.org/10.7554/eLife.33503.013

are to be able to stably induce a global perturbation in at least one grid module, and to do so at 2–3 different strengths. Critically, the perturbation must be one of the two specific types discussed above: a perturbation of the strength (gain) of inhibition in the network, or of the network time constants. The data to be collected are simultaneous recordings from several grid cells as the animal explores novel enclosures with no proximal spatial cues, over a $\geq 20$ minute trajectory.

First, before applying a perturbation, characterize spatial tuning (periods) and cell–cell relationships (relative spatial phases). Next, apply a series of 2–3 global perturbations of increasing strength. At each perturbation strength, characterize the spatial tuning of cells and cell-cell relationships.

A change in the amplitude of the grid cells' response across the different perturbations should signal that the perturbation is having an effect, regardless of underlying mechanism (**Figure 5**, first triangle on left).

If the different perturbation strengths do not cause a change in the spatial tuning periods of single cells (but the response amplitudes do change), it follows that velocity integration and spatial patterning are originating elsewhere, consistent with some feedforward mechanism (**Figure 5**, green).

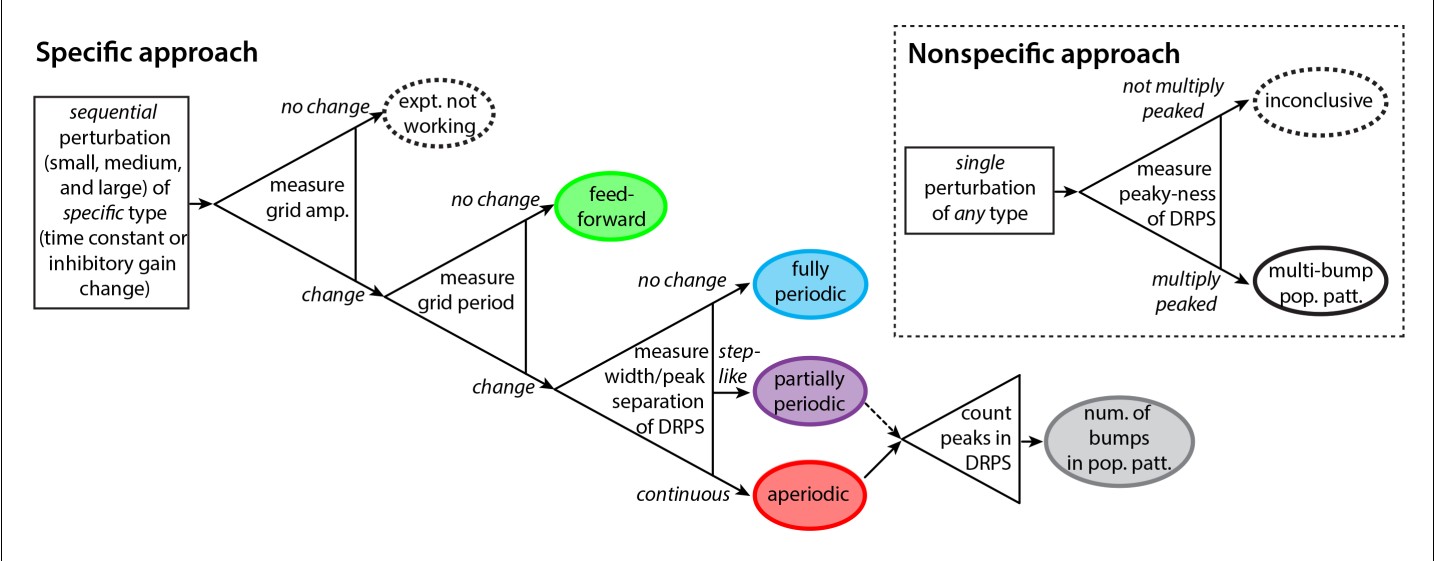

**Figure 5.** Decision tree for experimentally discriminating circuit mechanisms. The 'specific' approach involves a specific perturbation to either the gain of inhibition or the neural time-constants. Under the assumption of this kind of perturbation, the period, the amplitude, and the relative phases of the spatial tuning curves of neurons are measured pre-perturbation and then for each of three increasingly strong perturbations. A change in spatial tuning amplitude means that the attempted perturbation is in effect. Recurrent mechanisms can be discriminated from feedforward ones based on whether the perturbation changes the spatial tuning period (first open triangle). Different recurrent networks can be discriminated from each other based on the change in DRPS width or peak separation with perturbation strength (second open triangle). Finally, the number of bumps in the multi-bump population patterns can be inferred by counting the peaks in the DRPS (third open triangle), but for the partially periodic network only a lower bound on the number of bumps can be established (dotted line). Inset: 'Nonspecific' approach: After a perturbation of *any* type, the relative phases are measured. If the DRPS exhibits multiple peaks, then the underlying population pattern is multi-bump; otherwise, the test is inconclusive.

DOI: https://doi.org/10.7554/eLife.33503.014

The following figure supplements are available for figure 5:

**Figure supplement 1.** Perturbations applied to a random subset of neurons in the network.

DOI: https://doi.org/10.7554/eLife.33503.015

**Figure supplement 2.** Perturbations applied separately to the excitatory and inhibitory populations.

DOI: https://doi.org/10.7554/eLife.33503.016

**Figure supplement 3.** Perturbations applied to the gain of the neural response.

DOI: https://doi.org/10.7554/eLife.33503.017

**Figure supplement 4.** The effects of perturbations in networks with spatially untuned inhibitory neurons.

DOI: https://doi.org/10.7554/eLife.33503.018

**Figure supplement 5.** Learned place cell-based intrinsic error correction/resetting of grid phase in familiar environments is not predicted to play an important role in novel environments according to model.

DOI: https://doi.org/10.7554/eLife.33503.019

To confirm, verify that cell–cell relationships remain unchanged across perturbations, as also predicted for feedforward networks.

If there is a change in the spatial tuning period, characterize the cell–cell relationships in each perturbation condition. Plot the DRPS from each perturbed condition relative to the pre-perturbation condition, and quantify its width and if possible the separation between its peaks. If the DRPS width or peak separation increases steadily and smoothly with perturbation strength, that implies an aperiodic recurrent architecture (*Figure 5*, red). If the DRPS peak separation or width exhibits a step-like change, it is consistent with a partially periodic recurrent network (*Figure 5*, purple). Together with a change in the spatial tuning period, a DRPS that remains narrowly peaked at zero, with no change in width with perturbation strength, is consistent with a fully periodic network (*Figure 5*, blue).

Finally, if the network is either an aperiodic or partially periodic recurrent network, the number of peaks in the DRPS for each relative phase dimension is a lower bound on the quantity $2M$, where $M$ is the number of bumps in the population pattern along that dimension. If the stretch factor $\alpha$ times the number of bumps is smaller than 1/2 and the DRPS is multiply peaked the number of DRPS

peaks should equal twice the number of population activity bumps along the corresponding dimension (*Figure 5*, final triangle and gray oval).

The 'specific' approach above should provide insight into the underlying dynamics of the system with respect to the candidate models, regardless of outcome. By contrast, a 'nonspecific' approach (*Figure 5*, dashed box) could do the same, but only for certain outcomes. Suppose that after a number of any type of perturbations to the system, with known or unknown underlying mechanisms and at a local or systemic scale, one measures the DRPS. If the DRPS does not exhibit multiple peaks then, because this outcome is consistent with many possibilities and the nature of the perturbation is not precisely known or controlled to change the inhibitory gain or neural time-constant (the specific perturbations that provide higher discriminatory power), one cannot conclude anything about circuit architecture. On the other hand, if the DRPS after nonspecific perturbation does exhibit multiple, equi-spaced peaks, one can conclude with high confidence that the brain generates an underlying multi-bump population pattern through recurrent mechanisms with partially periodic or aperiodic structure. This is because a multi-peaked DRPS is a highly specific outcome of recurrent pattern-formation dynamics.

## Questions about experimental contingencies

- *Will it be possible to distinguish a 'no effect' result from an 'experiment is not working' result?* Specific inhibitory gain or time-constant perturbations are predicted to change the amplitude of the neural tuning curves relative to the unperturbed case in all models of *Figure 1*. An amplitude change from a specific perturbation is the signal that the experiment is working (*Figure 5*, first triangle on left).

- *If some circuit perturbation results in an amplitude change, can we learn something about the circuit from it?* If a perturbation affects response amplitudes without primarily affecting inhibitory gains or neural time constants, it qualifies as a 'nonspecific' perturbation. The ability to learn about circuit architecture then depends on the outcome: a multipeaked DRPS is informative, but a non-multipeaked DPRS is not (see *Figure 3* and *Figure 5* and above). For instance, direct perturbation of the amplitude of neural responses primarily through a change in the activation threshold of neurons (putatively the mechanism in *Kanter et al., 2017* through the action of DREADDs (*Sternson and Roth, 2014*; *Sánchez-Rodríguez et al., 2017*)) is predicted to result in amplitude changes but not a pattern period change in any of the candidate models (cf. *Figure 5—figure supplement 3*), and therefore cannot discriminate between feedforward and recurrent models unless the perturbation also affects gains or time constants. Computation of the DRPS would help resolve the question.

- *What if the perturbation targets only a (random) subset of grid cells in the circuit?* The qualitative predictions are unchanged if only a (random) subset of the neurons in the grid cell circuit are targeted by the perturbation. Quantitatively, the size of the effect (change in period for a given perturbation strength) will be scaled in recurrent network models by an amount proportional to the fraction, *Figure 5—figure supplement 1*.

- *What if the perturbation affects the time-constants of only excitatory (E) or only inhibitory (I) cells?* In an implementation of recurrent grid cell models with separate E and I populations (*Widloski and Fiete, 2014*), if the perturbation is specific to only E or only I cells (so long as the I cells are the ones that mediate inhibitory interactions between grid cells), the qualitative predictions regarding the direction of change in the spatial tuning period and in the DRPS remain unchanged (*Figure 5—figure supplement 2*).

- *What if the perturbation affects the gain of only E or only I cells?* The effect is the same as a time-constant change in the particular populations (see response above). On the other hand, changing the bias rather than the slope of the E cells, I cells, or both is a nonspecific perturbation (see question above); the population period in recurrent models is predicted to not vary with a bias change (*Figure 5—figure supplement 3*).

- *What if the perturbed population includes speed cells (Kropff et al., 2015) and pure head direction cells?* Changing the amplitude or time-constant of speed cells and head direction cells could further change the spatial tuning periods of grid cells by affecting the gain of velocity inputs to the system. However, perturbation of these cells should not by itself induce a shift in relative phases in recurrent and feedforward network models, so DRPS predictions remain as if the perturbed population did not include these cells.

- *What if the perturbed population includes conjunctive grid-head direction cells (Sargolini et al., 2006)?* The experiment would answer the question of whether grid activity patterns are generated through velocity integration by the targeted set of cells, without

distinguishing the relative contributions of these two populations. Narrowing the set of targeted cells to only one of these populations would provide finer-grained answers about mechanism. Similarly, if the experiment targets only grid cells and the mechanism is found to be feedfoward, the same experiments can be repeated one level upstream.

- *Would recurrent network models in which I cells are spatially untuned generate different predictions?* The model for spatially tuning in E cells and untuned I responses is in *Widloski and Fiete, 2014*. In simulation, the population period in the model can move in the opposite direction as a function of perturbation compared to models with spatially tuned I cells. However, the DRPS is sensitive to the magnitude but not the sign of phase shifts hence predictions about the DRPS remain qualitatively the same in both types of models (*Figure 5—figure supplement 4*).

- *What if errors in spatial phase estimation exceed the uncertainty assumed in this paper?* Most predictions (*Figure 3*) for differentiating between candidate mechanisms depend only on the DRPS envelope and its width and not on its detailed multi-peaked structure. Thus, they are fairly robust to phase estimation uncertainty (left two columns in (*Figure 4b*). Better phase estimation is only required if the goal is to determine the number of bumps in an underlying recurrently patterned network. As described earlier, this uncertainty can be reduced by averaging over longer trajectories.

- *Will it be possible to discover if grid cell responses are based on selective feedforward summation of the intermingled non-grid cells in MEC?* If the perturbation can be confined to exclude the intermingled non-grid cells in MEC, then it should be possible to use the perturbation and DRPS approach to tell apart a feedforward mechanism of spatial tuning inherited from non-grid MEC cells from a recurrent mechanism.

- *What if corrective inputs from place cells or external cues override the perturbation-induced change in the grid cell response?* This is a real possibility in familiar environments, and can mask a multi-peaked DRPS even if it would exist in the absence of corrective cues. For this reason, post-perturbation experiments should be performed in novel, featureless environments (*Figure 5—figure supplement 5*; more discussion below).

## Discussion

It is interesting to compare the potential of the present approach for discovering mechanism with other approaches. A high-quality, full-circuit connectome (*Seung, 2009*; *Briggman et al., 2011*) can specify the topology and locality of the network architecture. In other words, with appropriate analysis of the obtained data it should be possible to learn whether the connectivity matrix is 'local' (*Widloski and Fiete, 2014*) (*Figure 1a*), partially periodic (*Figure 1b*), or fully periodic (*Figure 1c*).

Network topology is, however, but one ingredient in circuit mechanism: Determining whether the observed connectivity actually accounts for the activity still requires inference (for instance, given a set of connections and weights, it is unknown whether they are strong enough to drive pattern formation in neural activity; determining this involves writing down a model of neural dynamics with the observed coupling). Even with further inference steps, whether the network originates certain functions like velocity-to-position integration and spatial tuning de novo (as in *Figure 1a–c*) or only amplifies or alters spatial tuning inherited from elsewhere (as in *Figure 1f*) cannot be answered by connectomics data. Despite their functional differences, feedforward and recurrent network models may exhibit similar lateral connectivity between grid cells. By contrast, the perturbative approach outlined here has the potential to reveal whether the function of path integration and spatial tuning originates in the perturbed set. The same approach can be sequentially applied to candidate areas progressively upstream of the grid cells.

Next, full-circuit activity data at single-neuron resolution can reveal much about the dynamics and dimensionality of the population response in the circuit. But without perturbation, inferring mechanism from activity alone is problematic: Materials and methods to estimate connectivity from activity (*Pillow et al., 2008*; *Roudi et al., 2009*; *Honey et al., 2009*) yield only effective couplings that reflect collective and externally driven correlations in addition to the true couplings. In other words, activity data alone without perturbation does not indicate where the observed activity arises or its mechanisms.

In summary, while connectomics and large-scale recording will provide vast amounts of valuable information, they are by themselves fundamentally correlative and thus not sufficient for discriminating between the candidate models discussed here. As we have shown, they may also not be

immediately necessary: a low-dimensional or 'global' perturbative approach which does not require targeting specific individual neurons according to their responses can yield rich information about mechanism, and can do so with a far sparser dataset.

Interestingly, cooling and similar perturbation experiments have been performed in V1 (*Michalski et al., 1993*; *Ferster and Miller, 2000*) but were not as revealing about underlying mechanism as they promise to be in grid cells. Why is this? Unfortunately, the candidate models of orientation tuning in V1 are ring networks (fully periodic, single-bump) or a feedforward mechanism, and as we have seen, these two models do not differ in their predictions for the DRPS (*Figure 3*). The multi-bump spatial tuning of grid cells derived from velocity integration at some stage offers a way to distinguish feedforward from recurrent models because perturbation at the integration stage is predicted produce a change in the spatial tuning curve period, an opportunity that does not exist in in V1. Thus, grid cells offer a unique opportunity to uncover the circuit mechanisms that support tuning curves and computation in the cortex, and our modeling work shows how to do so.

## Assumptions

We have assumed that the population pattern is stable against rotations (but the spatial tuning curves of cells are permitted to rotate) because a rotation would induce large changes in the DRPS and obscure the predicted effects of pattern expansion. Our assumption is supported by the observation that cell–cell phase relationships between grid cells are conserved across time and environments (*Yoon et al., 2013*), which can only hold if the underlying population pattern does not rotate.

The simplification that relative phases in the population pattern can be obtained from relative spatial tuning phases is valid if the intrinsically determined relative spatial phases of cells are not overridden by external spatial inputs. For instance, if an external cue (landmark or boundary) is associated with a specific configuration of grid cell phases, with the association acquired pre-perturbation, then the cue could activate the same configuration of grid cells post-perturbation, which can interfere with the perturbation-induced shifts in the intrinsic relative phases between these cells. To avoid this possibility it is important, post-perturbation, to assess spatial tuning relationships between cells only in novel environments, where there are no previously learned associations between external cues and the grid cell circuit. Ideally, these novel environments will be relatively free of spatial cues that resemble previously encountered cues and boundaries. Thus, the best environments for post-perturbation testing would be circular 2D arenas, differently colored, patterned, and scented, and with minimal distal cues beyond a global orienting cue; or virtual environments with visually textured but landmark-free walls (*Yoon et al., 2016*).

Even in novel environments, intrinsic error correcting mechanisms hypothesized in *Sreenivasan and Fiete, 2011* might trigger pre-perturbation grid cell configurations: a configuration of grid cells, after it is associated with a specific place field, can be triggered simply by activation of that place cell by another but similar grid cell configuration in the novel environment. We explore this possibility in a model and show that even after constructing associations of grid configurations with place fields at every location in two familiar environments, grid cell activations in a novel environment do not trigger activation of the learned place fields and their associated grid configurations from the familiar environments *Figure 5—figure supplement 5*. Based on this result, we believe that post-perturbation relative phases in grid cells may be relatively unaffected by intrinsic error-correction mechanisms in relatively featureless novel environments.

An interesting corollary to the possibility that previously learned reset or corrective inputs may co-activate cells that are out-of-phase cells post-perturbation (as is possible for partially periodic and aperiodic recurrent mechanisms) in familiar environments is that such resets should degrade rather than improve the quality of grid cell spatial tuning post-perturbation in previously learned environments.

Finally, it is important to note that if, in feedforward models, one were to include strong, continuous (rather than punctate, landmark-based) feedback from the grid cell layer to the spatially tuned inputs (as in *Bush and Burgess, 2014*), the network would effectively become a recurrent circuit that we have not included as a candidate. Similarly, we have excluded from our analysis recurrent network models of the spatial circuit with heterogeneous tuning and connectivity (*Cueva and Wei, 2018*; *Banino et al., 2018*; *Kanitscheider and Fiete, 2016*); these models do not yet capture the modular dynamics of the grid cell system, in which cells cluster in spatial period and those with similar period have the same orientation without the help of external aligning cues. When these models

are refined, and if the result is a distinct mechanism for modular grid cell dynamics than the candidate models considered here, it will be interesting to perform our proposed perturbations in them to obtain their predictions for experiment.

## Materials and methods

*Figure 1* and *Figure 1—figure supplement 1* are schematic. In *Figure 2*, *Figure 4a–b*, *Figure 2—figure supplement 2* and *Figure 4—figure supplement 1*, relative phase is computed from the population phases using idealized (hand-drawn) periodic population patterns that expand ($\delta^{ij} = \phi^i_{pop} - \phi^j_{pop}$), without the use of neural network simulations. *Figure 3*, *Figure 1—figure supplement 1*, *Figure 2—figure supplement 1*, *Figure 3—figure supplement 1*, Figure 2—figure supplement 4, *Figure 5—figure supplement 1*, *Figure 5—figure supplement 2*, *Figure 5—figure supplement 3*, and *Figure 5—figure supplement 4*, which distinguish between different recurrent architectures, are obtained by simulating the grid cell system in a neural network. Briefly, the network consists of excitatory and inhibitory neurons (except in *Figure 1—figure supplement 1* – see figure caption for details) with linear-nonlinear Poisson (LNP) spiking dynamics (*Burak and Fiete, 2009*; *Widloski and Fiete, 2014*) (except for *Figure 2—figure supplement 4*, where we use Hodgkin-Huxley dynamics). Structured lateral interactions between neurons pattern the neural population responses. Relative spatial tuning phases are computed from the tuning curves of different neurons, obtained by simulating the network response over 1 min long simulated quasi-random trajectories. The analysis of relative phase shifts, tuning amplitude and period in a network includes all cells with sufficiently good spatial tuning profiles: this set includes all cells in the fully and partially periodic networks and $3/4$ of the cells in aperiodic networks (from the central part of the network). Since the inhibitory and excitatory populations share similar population patterning and spatial tuning in these simulations (except in *Figure 5—figure supplement 4*), we arbitrarily display results from the inhibitory population.

### Neural network simulations

We use two different neuron models in our network simulations: LNP and Hodgkin-Huxley neurons, described below. Roman subscripts (e.g. $i, j$) refer to individual cells within population $P$. The population index $P$ can take the values $\{I, E^R, E^L\}$, designating inhibitory cells or excitatory cells that receive rightward or leftward velocity input, respectively. Integration is by the Euler method with time-step $dt$.

#### Linear-nonlinear-poisson (LNP) neurons

The time-varying firing rate $r^P_i(t) = f(G^P_i(t))$ of the $(P, i)$th cell is an instantaneous function of its time-varying summed input $G^P_i(t)$ with threshold-linear transfer function $f$:

$$f(x) = \begin{cases} x & x > 0 \\ 0 & x \leq 0. \end{cases} \tag{1}$$

Neurons emit spikes according to an inhomogeneous point process with rate $r^P_i(t)$ and coefficient of variance of CV = 0.5 (see *Burak and Fiete, 2009* and *Widloski and Fiete, 2014* for details on generating a sub-Poisson point process). LNP dynamics were used in all simulations except for *Figure 2—figure supplement 4*.

#### Cortical Hodgkin-Huxley (CHH) neurons

The membrane potential of the $(P, i)$th neuron is given by:

$$C_m \frac{dV^P_i}{dt} = -I^{ion,P}_i(V^P_i) - I^{syn,P}_i \tag{2}$$

where $C_m$ is the capacitance of the membrane, $I^{ion}(V)$ is the sum of the cell's intrinsic ionic currents, and $I^{syn}(V)$ is the current from recurrent and feedforward synaptic inputs to the cell. The ionic current is modeled as (*Pospischil et al., 2008*):

$$I^{ion}(V) = \overline{g}_L(V - \overline{V}_L) + \overline{g}_K n^4 (V - \overline{V}_K) + \overline{g}_M q(V - \overline{V}_K) + \overline{g}_{Na} m^3 h(V - \overline{V}_{Na}), \tag{3}$$

where the $\overline{g}$'s represent maximal conductance values and the $\overline{V}$'s are the reversal potentials of the leak conductance (L), the fast (K) and slow (M) potassium conductances, and the sodium conductance (Na). The dynamics and parameter settings of $n, m, q, h$ are as in *Pospischil et al., 2008* (we have replaced the 'p' gating variable in *Pospischil et al., 2008* with the notation 'q'). For CHH neurons, the time of a spike is defined as the time-step when the voltage crosses 0 mV from below. CHH dynamics were used in *Figure 2—figure supplement 4*.

## Synaptic activation

For both LNP and CHH neurons, spikes by the $(P, i)$th neuron activate all its outgoing synapses according to:

$$\frac{ds_i^P}{dt} + \frac{s_i^P}{\tau_{syn}} = \sum_b \delta(t - t_{i,b}^P), \tag{4}$$

where $t_{i,b}^P$ is the time of the $b$th spike and $\delta(t)$ is the Dirac delta function. The sum is over all spikes of the cell.

## Network inputs and interactions

We based our grid cell network models on the connectivity and weights that emerge from plasticity rules over a plausible developmental process, given in *Widloski and Fiete, 2014*, and thus might better represent the grid cell system than a model fully wired by hand. Moreover, the network contains both inhibitory and excitatory units (with the number of inhibitory units equalling 1/5 the number of excitatory units, like in cortex).

## Synaptic input to LNP cells

The total synaptic input $G_i^{syn,P}(t)$ into the $(P, i)$th LNP cell is given by

$$G_i^{syn,P} = [\alpha^{vel,P}(G_i^{rec,P} + G^0) + G^{0',P}]A_i^P, \tag{5}$$

where $\alpha^{vel,P}$ is the velocity input (described below), in multiplicative form; $G_i^{rec,P} = \sum_{P'} \sum_{j=1}^{N^{P'}} W_{ij}^{PP'} s_j^{P'}$ is the recurrent network input; $G^0, G^{0',P}$ are (small, positive) constant bias terms ($G^0 = 50, G^{0',I} = 0, G^{0',E^L} = G^{0',E^R} = 15$); and $A_i^P$ is a smooth envelope that modulates neural activity magnitudes across the network (described below).

To model *additive* velocity input, as in *Figure 2—figure supplement 1c*, we replace *Equation 5* with the following:

$$G_i^{syn,P} = [G^{vel,P} + G_i^{rec,P} + G^0 + G^{0',P}]A_i^P, \tag{6}$$

where $G^{vel,P} = W^{vel} \alpha^{vel,P}$ and $W^{vel} = 200$ ($\alpha^{vel,P}$ described below).

## Synaptic input to CHH cells

The total synaptic current $I_i^{syn,P}$ into the $(P, i)$th CHH neuron is given by

$$I_i^{syn,P} = \alpha^{vel,P} \left[ \sum_{P'} g_i^{rec,P} (V_i^P - \overline{V}^{P'}) + I^0 \right] A_i^P \tag{7}$$

where $\overline{V}^{P'}$ is the reversal potential for synaptic inputs from population $P'$ ($\overline{V}^E = 0$ mV and $\overline{V}^I = -80$ mV), $g_i^{rec,P} = \sum_j^{N^{P'}} W_{ij}^{PP'} s_j^{P'}$ is the recurrent network input, $I^0$ is a constant bias and $\alpha^{vel,P}, A_i^P$ are the same velocity and envelope terms mentioned above.

## Velocity input

The cells in the $P$th population receive a common motion-related input proportional to animal velocity along preferred direction $\hat{e}^P$:

$$\alpha^{vel,P} = 1 + \beta^{vel}\vec{v}\cdot\hat{e}^P, \tag{8}$$

where $\vec{v}$ is the instantaneous velocity of the animal and $\beta^{vel}$ is a scalar gain parameter. $\hat{e}^P = (0,0)$, $(0,1)$, $(0,-1)$ for the I, $E^R$ and $E^L$ populations, respectively. Unless otherwise noted, the velocity input is derived from a 1 min quasi-random trajectory (*Widloski and Fiete, 2014*).

## Recurrent weights

We based the recurrent weights $W_{ij}^{PP'}$ from cell $j$ in population $P'$ to $i$ in $P$ on those from the mature network of *Widloski and Fiete, 2014*.

We first describe weights in their *periodic* form. Let $x_{ij} = i - \gamma j$, where $\gamma = \frac{N_P}{N_{P'}}$ ($N_P$ is the number of neurons in population $P$). We also define the norm, $||x||_{N_P} \equiv \min(N_P - |x|, |x|)$. The E→I weights (i.e., $P = I$ and $P' = E^L, E^R$) are written as

$$W_{ij} = \frac{\eta}{\rho}\exp\left(\frac{-||x_{ij} - \rho\Delta||_{N_I}^2}{2(\sigma\rho)^2}\right), \tag{9}$$

where $\eta$ controls the overall weight strength, $\Delta$ and $\sigma$ control the shift and width, respectively, of the Gaussian profile, and $\rho$ is a scale factor that is used to shift from partially periodic ($\rho = 1$) to fully periodic ($\rho = 11$). The parameter $\rho$ takes the same values for the I-E and I-I weights (which are described below). The parameters are set as follows:

| Weight | $\eta$ | $\Delta$ | $\sigma$ |
|---|---|---|---|
| $E^L \to I$ | 11.5 | -2 | 4 |
| $E^R \to I$ | 11.5 | 2 | 4 |

The I→E weights are written as

$$W_{ij} = \frac{\eta}{\rho}\exp\left(\frac{-||x_{ij} - \rho\Delta||_{N_E}^2}{2(\sigma\rho)^2}\right)\Theta(||x_{ij}||_{N_E} - \rho\delta)\left[\Theta(-\mu x_{ij})\Theta(\mu x_{ij} + N_E/2) + \Theta(\mu x_{ij} - N_E/2)\right], \tag{10}$$

where $\Theta$ is the Heaviside function ($\Theta(x) = 1$ for $x >= 0$ and 0 otherwise). The first Heaviside function cuts out weights along the diagonal (the width of which is controlled by the parameter $\delta$), while the second, third, and fourth Heaviside functions together act as a windowing function to set to zero portions of the matrix to make the weights qualitatively resemble the developmental weights from *Widloski and Fiete, 2014*. The parameters are set as follows:

| Weight | $\eta$ | $\Delta$ | $\sigma$ | $\mu$ | $\delta$ |
|---|---|---|---|---|---|
| $I \to E^L$ | 4 | 8 | 10 | -1 | 3 |
| $I \to E^R$ | 4 | -8 | 10 | 1 | 3 |

Finally, the I→I weights are written as

$$W_{ij} = \frac{\eta}{\rho}\left[\exp\left(\frac{-||x_{ij} - \rho\Delta||_{N_I}^2}{2(\sigma\rho)^2}\right) + \exp\left(\frac{-||x_{ij} + \rho\Delta||_{N_I}^2}{2(\sigma\rho)^2}\right)\right]\Theta(||x_{ij}||_{N_I} - \rho\delta), \tag{11}$$

which is essentially a sum-of-Gaussians with the central portion removed (the width of which is controlled by $\delta$). The parameters are set as follows:

| Weight | $\eta$ | $\Delta$ | $\sigma$ | $\delta$ |
|---|---|---|---|---|
| $I \to I$ | 12 | 4 | 6 | 3 |

For the *aperiodic* network, the weights have the same form and parameter values as above (with $\rho = 1$), except with the following replacements:

$$|x| \leftarrow ||x||_N \tag{12}$$

$$A_{ij}^{PP'} W_{ij}^{PP'} \leftarrow W_{ij}^{PP'}. \tag{13}$$

where $|.|$ is the absolute value and $A_{ij}^{PP'} = A_i^P A_j^{P'}$ is an envelope function used to enforce a tapered profile on the weights, similar to **Burak and Fiete, 2009**:

$$A_i^P = \begin{cases} 1 & r_i^P < \kappa N_P \\ \exp\left[-a_0 \left(\frac{r_i^P - \kappa N_P}{(1-\kappa)N_P}\right)^2\right] & \text{otherwise,} \end{cases} \tag{14}$$

where $r_i^P = |i - \frac{N_P}{2}|$, and $\kappa = 0.3$ determines the range of the taper while $a_0 = 30$ controls its steepness.

## Changing pattern period by varying the 'neural' time-constant and the gain of recurrent inhibition in a network of LNP neurons

The period of the population pattern can be varied by rescaling the synaptic activation time constant, $\tau_{syn}$. It can also be varied by changing a gain parameter $\gamma_{inh}$ that controls the strength of synaptic weights from the inhibitory neurons: we set $W^{PI} \rightarrow \gamma_{inh} W^{PI}$, and allow $\gamma_{inh}$ to be varied away from unity.

The effect of time-constant on period in the different networks is quite non-trivial: It cannot be derived from a linear stability analysis on the network equations since it depends strongly on nonlinear interactions within the network bulk and with the network boundaries (**Widloski, 2015**). Instead, we study the effect though simulation of the nonlinear dynamics of the networks.

As noted in the main manuscript, neuromodulators can drive the requisite gain changes in recurrent weights. We show, through the more detailed Hodgkin-Huxley neuron simulations described below, that temperature may be used in experiments to cause similar changes in period as can be affected by changing recurrent weight strength, and that the effects of temperature change resemble the effects of changing the time-constant in the LNP model.

We study Hodgkin-Huxley (HH) dynamics to predict, with the help of more biophysically detailed neuron models and the documented variation of their parameters with temperature, the effects of cooling on population activity in grid cells. Specifically, we use a 'regular spiking' HH model of cortical neurons (**Pospischil et al., 2008**), which we supplement with models that describe temperature-induced changes in the parameters (**Hodgkin et al., 1952**; **Katz and Miledi, 1965**).

## Effects of temperature and neuromodulation on HH dynamics

Some HH models include modifications that capture the effects of temperature variation (**Hodgkin et al., 1952**; **Katz and Miledi, 1965**). These temperature effects are modeled by $Q_{10}$ factors that multiply the time-constants ($Q_{10}^{\tau} = 3$) and amplitudes ($Q_{10}^a = 1.3$) of the ionic conductances. At temperature $T$ (in °C), the conductance amplitudes $\overline{g}(T)$ and time constants $\tau(T)$ have the following form:

$$\overline{g}(T) \leftarrow \overline{g}(T_0)(Q_{10}^a)^{\frac{T-T_0}{10}} \tag{15}$$

$$\tau(T) \leftarrow \tau(T_0)/(Q_{10}^{\tau})^{\frac{T-T_0}{10}}. \tag{16}$$

where $T_0$ is 36°. We applied the $Q_{10}$ factor for $\overline{g}$ to the ionic conductance amplitudes $\overline{g}_L, \overline{g}_K, \overline{g}_M, \overline{g}_{Na}$ as well as to the synaptic conductance amplitudes $W_{ij}^{PP'}$. We also simultaneously applied the $Q_{10}$ factor for $\tau$ to the conductance and synaptic time-constants $\tau_n, \tau_q, \tau_m, \tau_h$ and $\tau_{syn}$. (For gating variable $x$, the time constant $\tau_x$ is defined as $\tau_x = 1/(\alpha_x + \beta_x)$, where $\alpha_x$ and $\beta_x$ are the rate constants governing the gating variable's dynamics (**Pospischil et al., 2008**).)

Finally, to isolate which parameters drove the strongest thermal effects on population patterning and the direction of these effects (so that we could extract lessons for how to vary parameters in grid cell models with simpler neuron dynamics) we applied thermal changes to the ionic conductances only (changing $\overline{g}_L, \overline{g}_K, \overline{g}_M, \overline{g}_{Na}, \tau_n, \tau_q, \tau_m, \tau_h$ according to the $Q_{10}$ factors while holding $W_{ij}^{PP'}$ and $\tau_{syn}$

constant), or to the synaptic conductances only (changing $W_{ij}^{PP'}$ and $\tau_{syn}$ according to the $Q_{10}$ factors while holding the ionic conductance parameters fixed).

To simulate the effects of a neuromodulatory gain change in inhibitory synapses, we set $W^{PI}$ to $\gamma_{inh} W^{PI}$, where $\gamma_{inh}$ is the prefactor modulating the strength of inhibition.

## Simulation parameters

### LNP dynamics

$N_{E_L} = N_{E_R} = 400$ neurons; $N_I = 160$ neurons; CV = 0.5; $dt = 0.5$ ms; $\tau_{syn} = 30$ ms*; $\beta^{vel} = 1$; $\gamma_{inh} = 1$*. (*: Indicates that parameters can change through perturbation.)

### Aperiodic network with CHH dynamics

All ionic conductance parameters are identical to those described in **Pospischil et al., 2008** for the RS model; as noted there, the parameters are set to values corresponding to a temperature of $T_0 = 36°C$. Synaptic weight definitions and parameter values same as LNP dynamics for aperiodic network (above), except that all $\eta$ values are scaled by the factor 0.0015 400 neurons; $N_I = 160$ neurons; $dt = 0.025$ ms; $\tau_{syn} = 15$ ms*; $\beta^{vel} = 0.8$; $C_m = 1$ $\mu$ F/cm$^2$; $\overline{g}_L = 0.1$ ms/cm$^2$*; $\overline{g}_K = 5$ ms/cm$^2$*; $\overline{g}_M = 0.07$ ms/cm$^2$*; $\overline{g}_{Na} = 50$ ms/cm$^2$*; $\overline{V}_L = -70$ mV; $\overline{V}_K = -90$ mV; $\overline{V}_{Na} = 50$ mV; $I^0 = 3$ $\mu$A/cm$^2$; $\rho = 1$. $\gamma_{inh} = 1$*. (*: Indicates that parameters can change through perturbation).

### LNP dynamics with E-E connections

All network parameters and synaptic weight definitions same as for LNP network (see above) with the addition of E-E connections (see below), except that the weight parameters have the following changes:

| Weight | $\eta$ | $\Delta$ | $\sigma$ | $\mu$ | $\delta$ |
|---|---|---|---|---|---|
| $E^L \rightarrow I$ | 3 | -2 | 8 | | |
| $E^R \rightarrow I$ | 3 | 2 | 8 | | |
| $I \rightarrow E^L$ | 3.25 | 8 | 8 | -1 | 3 |
| $I \rightarrow E^R$ | 3.25 | -8 | 8 | 1 | 3 |
| $I \rightarrow I$ | 4 | 4 | 6 | | 3 |

The E-E weights for the *periodic* networks are written similar to the E-I weights as

$$W_{ij} = \frac{\eta}{\rho} \exp\left( \frac{-||x_{ij} - \rho\Delta||_{N_E}^2}{2(\sigma\rho)^2} \right), \tag{17}$$

and have the following parameters:

| Weight | $\eta$ | $\Delta$ | $\sigma$ |
|---|---|---|---|
| $E^L \rightarrow E^L$ | 5.5 | -4 | 4 |
| $E^R \rightarrow E^R$ | 5.5 | 4 | 4 |
| $E^L \rightarrow E^R$ | 5.5 | 0 | 4 |
| $E^R \rightarrow E^L$ | 5.5 | 0 | 4 |

As in the LNP case, to get the *aperiodic* version of the E-E weights, replace

$$|x| \leftarrow ||x||_{N_E} \tag{18}$$

$$A_{ij}^{EE} W_{ij}^{EE} \leftarrow W_{ij}^{EE}. \tag{19}$$

where the envelope function $A_{ij}$ is described above.

## Alternative formulation of the DRPS

As before, the $i$th cell's firing phase within the periodic population activity pattern, defined as the cell's *population phase*, is $\phi_{pop}^i = ((i-1) \bmod \lambda_{pop})/\lambda_{pop}$ (with the arbitrary choice, made without loss of generality, that neuron 1 has phase 0) (*Figure 2—figure supplement 3b*, blue curve). For each cell in the population, plotting the pre-perturbation phase against the post-perturbation phase (red vs. blue curves in *Figure 2—figure supplement 3b*) shows that the data is quantized and lies on a series of parallel manifolds, *Figure 2—figure supplement 3c*. This quantization is captured via the following transformation to the phase shifts:

$$\Delta\phi_{pop}^i = \begin{cases} \phi_{pop,pre}^i - (1+\alpha)(\phi_{pop,post}^i - 1), & \text{if } \phi_{pop,pre}^i < (1+\alpha)\phi_{pop,post}^i \\ \phi_{pop,pre}^i - (1+\alpha)\phi_{pop,post}^i, & \text{otherwise,} \end{cases} \tag{20}$$

(we have assumed that the true stretch factor, $\alpha$, is known – later, we will show how $\alpha$ can be inferred from the data) followed by a modulo operation

$$\Delta\phi_{pop}^i = \Delta\phi_{pop}^i \bmod 1, \tag{21}$$

and then reflecting about the midpoint of the interval

$$\Delta\phi_{pop}^i = \min\{\Delta\phi_{pop}^i, 1 - \Delta\phi_{pop}^i\}. \tag{22}$$

The distribution of these phase shift values, *Figure 2—figure supplement 3d*, has three special properties: (1) The distribution is quantized, due to the fact that population activity pattern itself is quantized. (2) The number of peaks in the distribution is exactly equal to the number of bumps in the population activity pattern (this holds only for sufficiently small perturbations, such that $\Delta M < 0.5$, where $M$ is the number of bumps in the pre-perturbation population activity pattern – see *Figure 2—figure supplement 2* for explanation). (3) The peak separation in the distribution is exactly equal to the stretch factor, $\alpha$. The transformations described in *Equations 20-22* require knowledge of the stretch factor, $\alpha$, a quantity that is not directly observable. However, it can be inferred from the data, because the desired $\alpha$ value is the one that makes the distribution the most peak-y. This is equivalent to projecting the data onto its orthogonal axis, *Figure 2—figure supplement 3c*. Peaky-ness is quantified as the Pearson's correlation coefficient between the DRPS and a comb-like function defined over the same interval. The comb function is a series of delta-functions laid out with a spacing equal to $\alpha$. The desired $\alpha$ stretch factor is the one that maximizes this correlation (not shown).

## Correction of grid cell phases by grid cell-driven place cells

The model described below is based on work in *Sreenivasan and Fiete, 2011*. We assume $M$ modules, each with $N$ grid cells. The $i$th grid cell from the $m^{th}$ module in the $k^{th}$ environment has the following simplified tuning curve response:

$$f_{i,m,k}^{GC}(x) \propto \sin\left(\frac{2\pi x}{\lambda_m} + \phi_i + \widetilde{\phi}_{m,k}\right), \tag{23}$$

where $\lambda_m$ is the spatial period of the $m^{th}$ module, $\phi_i = 2\pi i/N$ is the cell's phase relative to others within the module (fixed across environments), and $\widetilde{\phi}_{m,k}$ is a random module-wide phase shift that is specific to each module and each environment, *Figure 5—figure supplement 5a*. The synaptic projections from grid cells to place cells are set as follows: Assume a population of $P$ place cells. For the $i$th place cell in the $k^{th}$ familiar environment, assign a random place preference, $x_{i,k}^{PC}$. The synaptic weight from the $(j,m)^{th}$ grid cell onto the $(i)^{th}$ place cell is incremented based on experience in environment $k$. The increment is Hebbian, given by the amplitude of the grid cell tuning curve at that place cell's preferred location:

$$\Delta W_{i,j,m}^k = f_{j,m,k}^{GC}\left(x_{i,k}^{PC}\right). \tag{24}$$

The total weight from grid cell $(j,m)$ to place cell $i$ is given by the sum of increments over all $L$ familiar environments:

$$W_{i,j,m} = \sum_{k=1}^{L} \Delta W_{i,j,m}^{k} = \sum_{k=1}^{L} f_{j,m,k}^{GC}\left(x_{i,k}^{PC}\right). \tag{25}$$

Given these weights, the $i^{th}$ place cell's full sub-threshold activity in environment $k$ is simply a weighted sum over the activities of the grid cells across modules, based on its weights:

$$f_{i,k}^{PC}(x) = \sum_{m}^{M} \sum_{j}^{N} W_{i,j,m} f_{j,m,k}^{GC}(x), \tag{26}$$

This description of place cell subthreshold activations holds for both familiar and novel environments; the only difference between familiar and novel environments is that in the latter there has been no increment of the grid cell-place cell weights based on coincident grid cell-place cell activity, **Figure 5—figure supplement 5a–b**. In the current implementation, we allowed every cell to have a field in every familiar environment. We see that even in this case, the subthreshold activations of PCs in the novel environment are far lower than at place fields in familiar environment; in other words, they will not be activated and drive correction or resetting of the grid cell phases in the novel environment. Including the measured degrees of sparseness in PCs should lead to even less interference than seen in simulated novel environment conditions.

## Measures used in main text
### DRPS in 1D
The relative phase of cell $i$ and $j$ is defined as $\delta^{ij} = d^{ij} \bmod \lambda/\lambda$, where $d^{ij}$ is the offset in the central peak of the cross-correlation in their spatial tuning curves, and $\lambda$ is their common spatial period (in the main text, for **Figures 2**, **4a-b**, **Figure 2—figure supplement 2**, and **Figure 4—figure supplement 1**, the relative phase is computed directly from the population phases, that is, $\delta^{ij} = \phi_{pop}^{i} - \phi_{pop}^{j}$. The relative phase magnitude is given by $|\delta| = \min(||\delta||, 1 - ||\delta||)$, where $||\cdot||$ is the absolute value norm. The DRPS is computed by making a distribution of phase magnitude shifts, $|\delta_{pre}| - |\delta_{post}|$, where $\delta_{pre}$ and $\delta_{post}$ are the relative phases measured pre- and post-perturbation.

### 2D relative phase
For two cells $i$ and $j$, let $\vec{d}$ be the displacement vector which measures the 2D offset in the central peak of the cross-correlation in their spatial tuning curves. The displacement vector is converted into a 2D phase $\vec{\delta}$ according to $\vec{\delta} = (\delta_1, \delta_2) = f(d_1^{proj}/\lambda_1 \bmod 1, d_2^{proj}/\lambda_2 \bmod 1)$, where $\vec{d}^{proj} = (d_1^{proj}, d_2^{proj})$ is the oblique projection of $\vec{d}$ onto the principal vectors $\lambda_1 \hat{e}_1$ and $\lambda_2 \hat{e}_2$, and

$$f(\vec{x}) = \begin{cases} (x_1 - 1, x_2 - 1) & \text{if } x_1 \geq 0.5 \text{ and } x_2 \geq 0.5 \\ (x_1 - 1, x_2) & \text{if } x_1 \geq 0.5 \text{ and } x_2 < 0.5 \\ (x_1, x_2 - 1) & \text{if } x_1 < 0.5 \text{ and } x_2 \geq 0.5 \\ (x_1, x_2) & \text{if } x_1 < 0.5 \text{ and } x_2 < 0.5. \end{cases} \tag{27}$$

### DRPS in 2D
The DRPS in 2D is computed separately for the two components of the 2D relative phase. That is, given the relative phase vector $\vec{\delta} = (\delta_1, \delta_2)$, the DRPS is computed by making a distribution of phase magnitude shifts for each component: $|\delta_{1,pre}| - |\delta_{1,post}|$ and $|\delta_{2,pre}| - |\delta_{2,post}|$, where the magnitude is defined as the absolute value norm: $|\cdot| = ||\cdot||$.

### Bootstrap resampling and phase uncertainty
Given an original spike map of $M$ total spikes (with locations) from one cell, we created a new spike map of $N$ ($N < M$) total spikes, by picking spikes (with their corresponding location coordinates) from the original map one at a time, at random, and with replacement. The same was done for a second, simultaneously recorded cell. From these sampled spike trains for a pair of cells, we estimated relative phase (by computing the location of the peak closest to the origin in the cross-correlation of the spatial maps of the two cells, as in **Yoon et al., 2013**). The procedure was performed 100 times, generating 100 bootstrapped relative phase estimates per cell pair. Phase uncertainty was measured

as the peak location of the Rayleigh distribution that best fit the distribution of magnitudes of the bootstrapped relative phase estimates.

## Spatial tuning curves

For a given cell and trajectory, we build a histogram of spike counts at each location (bin size = 1 cm), then normalize the count in each bin by the amount of time spent in it. The normalized histogram is smoothed by convolution with a boxcar filter (width = 5 bins) to yield a spatial tuning curve.

## Spatial tuning period and amplitude

The spatial tuning period is measured as the inverse of the spatial frequency with the highest peak in the power spectrum of the spatial tuning curve (excluding the peak at 0 frequency). Likewise, the spatial tuning amplitude is measured as the mean spike rate density across the bins of the spatial tuning curve. The quantities reported in *Figure 3* and *Figure 3—figure supplement 1* are averaged over all cells in the population.

## Population activity period and gridness

The population activity gridness is taken to be the power of the largest frequency component of the power spectrum measured from a normalized snapshot (frame) of the population activity (normalized = mean subtracted, followed by division by standard deviation). The power spectrum is rescaled by the factor $2/L^2$, where L is the number of bins in the population activity vector from which the power spectrum was computed. The population activity vector is shortened to include only the middle one-half of the population, so that for the $E^L$ population, L is 100. From the power spectrum, the population activity period is taken to be the wavelength at which the power spectrum has the largest peak. Throughout the paper, both the reported population period and gridness are averaged over the last 10000 snapshots of the population activity pattern from a given trial.

## Velocity response

Velocity response is measured as the translation speed (neurons/sec) of the network pattern to fixed input velocity, computed by tracking the displacement of the pattern for 10 s, smoothing the resulting trajectory with an 4 s moving average filter, and then measuring the average speed of the middle-half of the trajectory.

## Periodicity score for the DRPS

We smooth the histogram of relative phase shifts (by convolution with a 2-bin Gaussian kernel) and normalize it (by mean subtraction and division by the standard deviation). Next, we compute the power spectrum, rescaling the result by $2/L^2$, where $L$ is the number of bins in the histogram ($L = 200$). The periodicity score is set to be the power of the largest- amplitude non-zero frequency component in the scaled power spectrum. This score returns 1 if the DRPS is a pure sinusoid. It returns 0 if the DRPS is flat and returns an average value of $<0.2$ if the DRPS were constructed bin by bin by taking independent, identically distributed (iid) samples from a uniform distribution on the unit interval.

## Acknowledgments

We are grateful to Dori Derdikman, Caswell Barry, Laura Colgin, and Lisa Giocomo for useful discussions and Rishidev Chaudhuri and Ingmar Kanitscheider for comments on the manuscript. This work was supported in part by grants from the Human Frontiers in Science Program (HFSP-RGP0062/2014), the National Science Foundation (NSF-CRCNS- IIS-1311213), the Howard Hughes Medical Institute through the Faculty Scholars Program, and the Simons Foundation through the Simons Collaboration on the Global Brain.

## Additional information

### Funding

| Funder | Grant reference number | Author |
|---|---|---|
| Human Frontier Science Program | HFSP-RGP0062/2014 | Ila R Fiete |
| National Science Foundation | NSF-CRCNS- IIS-1311213 | Ila R Fiete |
| Howard Hughes Medical Institute | | Ila R Fiete |
| Simons Foundation | | Ila R Fiete |

The funders had no role in study design, data collection and interpretation, or the decision to submit the work for publication.

### Author contributions

John Widloski, Conceptualization, Formal analysis, Investigation, Methodology, Writing—original draft, Writing—review and editing; Michael P Marder, Formal analysis, Investigation, Methodology; Ila R Fiete, Conceptualization, Supervision, Funding acquisition, Methodology, Writing—original draft, Writing—review and editing

### Author ORCIDs

John Widloski (iD) https://orcid.org/0000-0003-4236-8957
Ila R Fiete (iD) https://orcid.org/0000-0003-4738-2539

### Decision letter and Author response

Decision letter https://doi.org/10.7554/eLife.33503.022
Author response https://doi.org/10.7554/eLife.33503.023

## Additional files

### Supplementary files

• Transparent reporting form
DOI: https://doi.org/10.7554/eLife.33503.020

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
