## [Decision Letter]

Thank you for submitting your article "Inferring circuit mechanisms from sparse neural recording and global perturbation in grid cells" for consideration by *eLife*. Your article has been reviewed by three peer reviewers, and the evaluation has been overseen by a Reviewing Editor and Michael Frank as the Senior Editor. The following individual involved in review of your submission has agreed to reveal their identity: Omri Barak (Reviewer #3).

The reviewers have discussed the reviews with one another and the Reviewing Editor has drafted this decision to help you prepare a revised submission.

Summary:

In the article "Inferring circuit mechanisms from sparse neural recording and global perturbation in grid cells" the authors propose a way to show how different models of grid cell activity can be distinguished/falsified through experiments both in silico and in vivo. A wide variety of models explaining the grid cell pattern and network have been put forward since the discovery of grid cells in 2005. According to the authors existing data cannot distinguish between recurrent and feedforward models. The paper especially emphasizes velocity-based, continuous attractor models of grid cells using recordings from a small selection of neurons and methods to perturb the system. The authors claim that their suggested perturbation (cooling) combined with their novel measure (DRPS) could in part solve the "inverse problem" of inferring circuitry. The authors make the distinction between models that assume 1) a connectivity profile where the connectivity depends on the phase relationship between the neurons, termed the "fully connected", and those where the connectivity is determined by a distance between the neurons either 2) on a torus, "partially periodic", or 3) on a plane with tweaks on the edges, "aperiodic". By manipulating the network (e.g. changing the strength of the inhibition), the authors demonstrate that there can be a detectable change in the phase relationship between neurons in the partially periodic and aperiodic models. The manuscript offers quite an interesting and unusual perspective by giving direct suggestions on an experiment that could distinguish which of the models are correct. The last figure of the paper (Figure 5) is actually a very pedagogical decision tree for experimentally discriminating the underlying circuit mechanisms.

The paper is on one hand of great appeal, but on the other hand has major flaws: On one hand it is of great interest to the community, due to its nice link between a whole set of theories and potential experiments. Beyond the clear appeal to people working on grid cells, the systematic treatment of perturbation-based predictions could be relevant to other realms as well. However, all the reviewers have pointed to serious flaws in the paper, which should be addressed in order for the paper to be reconsidered for publication in *eLife*.

Given our concerns, we ask that you respond soon with a detailed plan to address the essential points below and provide an estimate of the time it may take to do so. The reviewing editor and referees will then consider your proposed work and issue a binding recommendation

Essential revisions:

1) Connection to real data: The paper would have been much stronger if it would have been more connected to real experiments. We note that an experiment including recordings of grid cells during a perturbation has already been published (Kanter et al., 2017). The authors in that paper used chemogenetic manipulations to determine the effects of hyperpolarization and depolarization on spatial coding in grid cells and how that is later read out by hippocampal place cells. Both manipulations resulted in reliable changes to the spatial tuning amplitudes but without a change in the placement of the fields in the environment or in the phase relationship between cells. We believe that the article here would suggest that this is the result of a feedforward mechanism and that the DRPS method would not be applicable. It seems appropriate that the authors mention this work and discuss under what conditions they might expect a different result.

2) Accounting for additional phenomena in grid cells: The method described here is designed to discriminate mostly between variants of the purely velocity-driven continuous attractor models of single modules. While these models have been useful in demonstrating how a network could integrate velocity, it is well known that any deviations from the assumed connectivity will cause the network to drift (Tsodyks and Sejnowski, 1995; Zhang, 1996), resulting in significant errors over time. Of course, this is easily remedied by additional spatial inputs (e.g. Pastoll et al., 2013) that could come from cues in the environment, such as encounters with a wall, or other mechanisms such as a very appealing interaction of grid modules of different spacing and the place cell system, as suggested by Sreenivasan and Fiete, 2011.

Furthermore, there is additional experimental work that single module velocity-driven networks would likely be insufficient to replicate, such as the field placements in a trapezoid (Krupic et al., 2015), the distortions (Stensola et al., 2015) and field-to-field variability in boxes (Dunn et al., 2017; Ismakov et al., 2017; Kanter et al., 2017).

The authors have to think whether the DRPS method would still be effective if drift or any of these experimentally-observed details are accounted for.

3) Scaling of velocity inputs: The model described here is a variant of the model described in Widloski and Fiete, 2014. This model multiplies the velocity input (Equation 7) by the synaptic input (see Equations 5 and 6). This is in contrast to the additive velocity input in Burak and Fiete, 2009 as well as many other models. This is an important detail since scaling the velocity input by the synaptic input partially mitigates the issue of the balance between the velocity and non-velocity input. It might be that if the velocity input was included additively, the network would much sooner result in extremely large or small periods than any detectable differences in phase relationships. Thus, the authors should check this more plausible variant of the model.

4) The scope of the models: The results of the paper are only valid within a limited set of models, that is not as inclusive as the authors describe. The authors present their analysis as a complete survey of all plausible candidate models. However, there are many other options. For instance, a feed forward model that also has strong lateral connections in the grid layer is one example of a hybrid model. We don't think the authors should cover every conceivable model, but the scope of the study could be stated more clearly.

Furthermore, Figure 5 suggests that, after perturbation, if the grid period has not changed then it should be a feedforward network. It would be more correct to say that it would rule out a purely velocity-driven, single-module continuous attractor model. One example of a non-feedforward mechanism that includes continuous attractor dynamics and might be able to handle (to some degree) such a perturbation would be Sreenivasan and Fiete, 2011. According to Figure 5, the experiment does not work if no change is observed, but maybe it is rather the model that were incorrect?

Related to the above points are the results presented in Figure 5—figure supplement 1. Does this indicate that not all the leaves in Figure 5 are expected to exist? In general, how robust are the different model classes?

We understand that the authors have to limit their investigation to a limited number of grid cell models, but it should be more clearly stated that the suggested experiment can only discriminate between these attractor dynamic models. Furthermore, a short discussion of grid cell models based on other principles should be mentioned in the manuscript.

5) Biological complexity: The authors suggest another experiment, manipulating the gain of inhibitory synaptic conductances, for example by infusion of benzodiazepines. Furthermore, they state that this manipulation has "unambiguous interpretation in terms of grid cell models". This statement is too strong, as we assume that the authors are not ignorant to the complexity of biological systems and cortical networks. None of the models considered in the paper takes into account the different cell types or different connectivity among the neurons in the different layers of entorhinal cortex (e.g. Fuchs et al., 2016). Without speculating on the effects of infusing a drug, it is likely not as clean as adjusting all the synaptic weights to the same amount in a model.

Furthermore, while the method described in the manuscript are able to infer circuitry and mechanism from a sparse population of "recorded" neurons, it does not seem to us that the authors consider that the model neurons are very homogeneous, in contrast to all in vivo recordings, that contain a lot of neurons that vary in tuning curves, regularity, firing rates, grid scores etc. Since the aim of the paper is to infer connectivity, there should be a discussion related to the different cell types and connections in the entorhinal network. We do not expect the authors to build a model representing all the biological complexity of different cell types, connections etc., but we think the paper would have improved if the authors discussed to what extent their model is robust to biological complexity.

[Editors' note: the authors’ plan for revisions was approved and the authors made a formal revised submission.]

---

## [Author Response]

Essential revisions:1) Connection to real data: The paper would have been much stronger if it would have been more connected to real experiments. We note that an experiment including recordings of grid cells during a perturbation has already been published (Kanter et al., 2017). The authors in that paper used chemogenetic manipulations to determine the effects of hyperpolarization and depolarization on spatial coding in grid cells and how that is later read out by hippocampal place cells. Both manipulations resulted in reliable changes to the spatial tuning amplitudes but without a change in the placement of the fields in the environment or in the phase relationship between cells. We believe that the article here would suggest that this is the result of a feedforward mechanism and that the DRPS method would not be applicable. It seems appropriate that the authors mention this work and discuss under what conditions they might expect a different result.

Thank you for the comment, it is indeed valuable to address what can be learned about circuit mechanism from perturbations that might differ from what we have proposed, and to address how existing perturbation studies relate to our predictions. In response, we have added next to the manuscript, discussing a new experimental condition, called "nonspecific perturbation". We discuss how Kanter et al. qualifies as a nonspecific perturbation according to the definition. We have also summarized what can be learned from "nonspecific perturbations" to the experimental workflow diagram of Figure (Figure 5). We also now consider the Kanter et al. 2017 study in the section "Questions about experimental contingencies".

To summarize the content: The layer-II specific perturbation in Kanter is an induction of a rate change, presumably/putatively through a change in threshold through depolarizing or hyperpolarizing drive to the MEC stellate cells via DREADDs***. In our new terminology, if the primary perturbation mechanism is not through a change in neural time constant or inhibitory gain, then it is a "nonspecific" perturbation. Nonspecific perturbations can shed light on mechanism if the DRPS is computed and found to be multiply peaked, but a nonspecific perturbation result is not informative about feedforward versus recurrent mechanisms and about different varieties of recurrent mechanisms if the DRPS is not multiply peaked.

According to Figure 5—figure supplement 3, driving a rate change directly should not result in a change in period for any network connectivity (even though it is expected to result in an amplitude change, as seen in the experiment). Thus, this is not the type of perturbation that is informative about network configuration.

*** The perturbation is by CNO action through GIRK and phospholipase-C; a recent study (Sanchez-Rodriguez et al. Sci. Reports 2017; doi: 10.1038/s41598-017-15306-8) suggests that GIRK does not affect the I-O curve of the target neurons except by a small shift (suggesting a threshold shift rather than a gain change), consistent with other studies (Sternson et al., 2014).

2) Accounting for additional phenomena in grid cells: The method described here is designed to discriminate mostly between variants of the purely velocity-driven continuous attractor models of single modules. While these models have been useful in demonstrating how a network could integrate velocity, it is well known that any deviations from the assumed connectivity will cause the network to drift (Tsodyks and Sejnowski, 1995; Zhang, 1996), resulting in significant errors over time. Of course, this is easily remedied by additional spatial inputs (e.g. Pastoll et al., 2013) that could come from cues in the environment, such as encounters with a wall, or other mechanisms such as a very appealing interaction of grid modules of different spacing and the place cell system, as suggested by Sreenivasan and Fiete, 2011.Furthermore, there is additional experimental work that single module velocity-driven networks would likely be insufficient to replicate, such as the field placements in a trapezoid (Krupic et al., 2015), the distortions (Stensola et al., 2015) and field-to-field variability in boxes (Dunn et al., 2017; Ismakov et al., 2017; Kanter et al., 2017).The authors have to think whether the DRPS method would still be effective if drift or any of these experimentally-observed details are accounted for.

This is an excellent question. Please note that a drift in the phase of the active grid cells relative to the integrated velocity input and therefore relative to the actual position of the animal itself is not in itself a problem for our method, because this work is concerned with relative phases between pairs of cells rather than the absolute phase of cells. Relative phase can be accurately estimated if it is measured in a time-resolved way using small-time windows (as done for instance in Hafting et al. 2005 or Yoon et al. 2013).

However, as the reviewers and our original manuscript both note, drift might introduce a problem arising from the corrective mechanisms the system might implement to counteract it (e.g. correction of grid phases by another cell type like border or place cells, based on previously learned cue-to-grid phase associations). The corrective inputs, by enforcing patterns of grid phase consistent with pre-perturbation learning, might mask the effects of the perturbation on the population period. For this reason, we proposed that the experiment be performed in novel environments, where we assumed the corrective inputs would be minimal. However, this was an assumption that we did not test by modeling. We now explicitly build a model with corrective feedback to show that the corrective feedback wired up from previously encountered environments need not seriously interfere with model predictions in novel environments.

Specifically, we have thus built a model, similar to Sreenivasan and Fiete 2011, in which particular grid cell phase combinations are associated with a particular place cells. A grid cell pattern similar but not identical to one of these combinations would activate one of the place cells, which would in turn activate the previously learned grid cell phase combination. In the familiar environment, this would serve as a corrective mechanism. However, in a novel environment, the same mechanism might also enforce previous grid cell combinations, thus rendering any perturbation ineffective. In the model, we find that in novel environments, where grid cell phases are initialized randomly, previously learned grid cell-place cell combinations do not rise above the threshold for place cell activation, and thus do not trigger the corrective mechanism(see Figure 5—figure supplement 5). This is also plausible for and consistent with the actual biological scenario, because if the corrective mechanisms from familiar environments were continuously triggered in novel environments, there would be no remapping.

3) Scaling of velocity inputs: The model described here is a variant of the model described in Widloski and Fiete, 2014. This model multiplies the velocity input (Equation 7) by the synaptic input (see Equations 5 and 6). This is in contrast to the additive velocity input in Burak and Fiete, 2009 as well as many other models. This is an important detail since scaling the velocity input by the synaptic input partially mitigates the issue of the balance between the velocity and non-velocity input. It might be that if the velocity input was included additively, the network would much sooner result in extremely large or small periods than any detectable differences in phase relationships. Thus, the authors should check this more plausible variant of the model.

We have now performed simulations for both the additive and multiplicative velocity models; the results are qualitatively unchanged (Figure 2—figure supplement 1C).

4) The scope of the models: The results of the paper are only valid within a limited set of models, that is not as inclusive as the authors describe. The authors present their analysis as a complete survey of all plausible candidate models. However, there are many other options. For instance, a feed forward model that also has strong lateral connections in the grid layer is one example of a hybrid model. We don't think the authors should cover every conceivable model, but the scope of the study could be stated more clearly.Furthermore, Figure 5 suggests that, after perturbation, if the grid period has not changed then it should be a feedforward network. It would be more correct to say that it would rule out a purely velocity-driven, single-module continuous attractor model. One example of a non-feedforward mechanism that includes continuous attractor dynamics and might be able to handle (to some degree) such a perturbation would be Sreenivasan and Fiete, 2011. According to Figure 5, the experiment does not work if no change is observed, but maybe it is rather the model that were incorrect?Related to the above points are the results presented in Figure 5—figure supplement 1. Does this indicate that not all the leaves in Figure 5 are expected to exist? In general, how robust are the different model classes?We understand that the authors have to limit their investigation to a limited number of grid cell models, but it should be more clearly stated that the suggested experiment can only discriminate between these attractor dynamic models. Furthermore, a short discussion of grid cell models based on other principles should be mentioned in the manuscript.

We have edited the Introduction and Results to make clear the scope/range of models that we are proposing to distinguish between. We clarify in the caption, Results and the Discussion that when we talk about "ruling out" models, we are referring to the specific candidate models identified in the manuscript; we have also clarified what candidate models we are considering.

In the Discussion, we already had already mentioned that hybrid models with strong feedforward and feedback mechanisms are not addressed;we have now expanded the paragraph in the Discussion on models that are not included as candidate models for this study (last paragraph), including feedforward models with strong lateral connectivity and neural network models trained to have strong recurrent weights that are heterogenous, with heterogeneous neural tuning.

Next, adding the correction mechanism of Sreenivasan and Fiete 2011 should not cause the network to resist the effects of perturbation in novel environments: Please see our response to Essential Revision 2) above. Finally, please note that the measure of the perturbation not working is if there is no change in amplitude (as indicated in Figure 5), a change in amplitude is predicted as a response to the perturbation for all the candidate models. A non-change or change in period then differentiates purely feedforward from purely recurrent models. Indeed, we cannot exclude the possibility that there are other models which might have different predictions; however, the scope of this work, which we believe already carries high value, is to discriminate among the set of candidate models we specifically define at the beginning of the paper. (Any of the candidate models could exist; we simply believe some of the candidate models have a smaller prior probability than others. Figure 1—figure supplement 2 relates to these prior probabilities.)

5) Biological complexity: The authors suggest another experiment, manipulating the gain of inhibitory synaptic conductances, for example by infusion of benzodiazepines. Furthermore, they state that this manipulation has "unambiguous interpretation in terms of grid cell models". This statement is too strong, as we assume that the authors are not ignorant to the complexity of biological systems and cortical networks. None of the models considered in the paper takes into account the different cell types or different connectivity among the neurons in the different layers of entorhinal cortex (e.g. Fuchs et al., 2016). Without speculating on the effects of infusing a drug, it is likely not as clean as adjusting all the synaptic weights to the same amount in a model.

Indeed, thank you for pointing this out, it was a phrasing error: We meant that "within the models, an inhibitory gain change has an unambiguous interpretation", not that the effects of benzodiazepines would unambiguously be a clean gain change and nothing else. As the reviewers note, infusion of a drug is not likely to result in a single, clean effect such as adjusting all synaptic weights. (Nevertheless, the model is "robust" in the sense that if there is a net gain change effect, we do expect the qualitative predictions to still hold.) We have edited the text to clarify what was meant.

With respect to the complexity of predicting drug-based responses and the possibility that different subpopulations might be affected differentially, the section "Questions about experimental contingencies" addresses some possibilities about what to expect if only subsets of the neurons of specific types are perturbed.

Furthermore, while the method described in the manuscript are able to infer circuitry and mechanism from a sparse population of "recorded" neurons, it does not seem to us that the authors consider that the model neurons are very homogeneous, in contrast to all in vivo recordings, that contain a lot of neurons that vary in tuning curves, regularity, firing rates, grid scores etc. Since the aim of the paper is to infer connectivity, there should be a discussion related to the different cell types and connections in the entorhinal network. We do not expect the authors to build a model representing all the biological complexity of different cell types, connections etc., but we think the paper would have improved if the authors discussed to what extent their model is robust to biological complexity.

As suggested, we now add a Discussion paragraph about models that exhibit more complex responses or network architecture (please see last paragraph, Discussion).

Briefly, it is true that cells near and around grid cells in the entorhinal cortex are heterogeneously tuned. However, from the experimental perspective, it is not yet known whether these different cells together form a strongly interconnected circuit through a different mechanism than the candidate models considered here. From the modeling perspective, even the most recent developments from training (LSTM) neural networks to obtain grid and other spatial cell types do not result in grid cells cell populations with near-identical period and aligned orientation, without external environmental aligning cues, leaving open the question of whether it is possible to construct qualitatively different models from the existing ones that still reproduce much grid cell phenomenology. Nevertheless, when the models are refined, and if they produce modular grid cell responses, it will indeed be interesting to apply the proposed perturbations to them to obtain predictions for experiment.

We have added a question related to the comment above under "Questions about experimental contingencies". The question reads: "Will it be possible to discover if grid cell responses are based on selective feedforward summation of the intermingled non-grid cells in MEC?"